# Importance of natural land cover for plant species' conservation: A nationwide study in The Netherlands

**Kaixuan Pan**[1]*, **Merijn Moens**[1,2], **Leon Marshall**[2,3], **Ellen Cieraad**[1], **Geert R. de Snoo**[1,4], **Koos Biesmeijer**[1,2]

**1** Institute of Environmental Sciences, Leiden University, Leiden, The Netherlands, **2** Naturalis Biodiversity Center, Leiden, The Netherlands, **3** Agroecology Lab, Université Libre de Bruxelles (ULB), Brussels, Belgium, **4** Netherlands Institute of Ecology, Wageningen, The Netherlands

\* k.pan@cml.leidenuniv.nl

## Abstract

While shifts to high-intensity land cover have caused overwhelming biodiversity loss, it remains unclear how important natural land cover is to the occurrence, and thus the conservation, of different species groups. We used over 4 million plant species' observations to evaluate the conservation importance of natural land cover by its association with the occurrence probability of 1 122 native and 403 exotic plant species at 1 km resolution by species distribution models. We found that 74.9% of native species, 83.9% of the threatened species and 77.1% rare species preferred landscapes with over 50% natural land cover, while these landscapes only accounted for 15.6% of all grids. Most species preferred natural areas with a mixture of forest and open areas rather than areas with completely open or forested nature. Compared to native species, exotic species preferred areas with lower natural land cover and the cover of natural open area, but they both preferred extremely high and low cover of natural forest area. Threatened and rare species preferred higher natural land cover, either cover of natural forest area or cover of natural open area than not threatened and common species, but rare species were also more likely to occur in landscapes with 0–25% cover of natural open area. Although more natural land cover in a landscape will not automatically result in more native species, because there is often a non-linear increase in species occurrence probability when going from 0% to 100% natural land cover, for conserving purposes, over 80% natural land cover should be kept in landscapes for conserving threatened and very rare species, and 60% natural land cover is the best for conserving common native species. Our results stress the importance of natural areas for plant species' conservation. It also informs improvements to species conservation by increasing habitat diversity.

**Data Availability Statement:** The dataset generated for supporting the analysis in this study is included in the Supplementary Information. The

raw data of species occurrence are available from the Dutch National Database of Flora and Fauna (NDFF, www.ndff.nl) but restrictions apply to the availability of these data, which were used under license for the current study, and so are not shared publicly. This restriction has been imposed by the Dutch National Database of Flora and Fauna. However, these data are available upon valid request from NDFF service team, serviceteamNDFF@natuurloket.nl. In fact, Dutch government is currently moving towards making the NDFF data publicly available.

**Funding:** K. P. received the funding from the China Scholarship Council (Grant No. 201806320120). URL: https://www.chinesescholarshipcouncil.com/. L. M. was supported by the 'Fonds Wetenschappelijk Onderzeok – Vlaanderen (FWO)' under EOS Project (no. 3094785). URL: https://www.fwo.be/. The funders had no role in study design, data collection and analysis, decision to publish, or preparation of the manuscript.

**Competing interests:** The authors have declared that no competing interests exist.

## Introduction

Global biodiversity is declining at an unprecedented rate and around one million plant and animal species are facing extinction [1, 2]. The transformation of natural habitat into agriculture, infrastructure and urban areas is considered as one of the main drivers of these declines [3, 4]. Currently, approximate three-quarters of the terrestrial biosphere has been altered by human activities [5] and the impact of these alterations on biodiversity is expected to increase this century [6]. Natural (e.g. natural forest, wetland and dunes, *etc*.) and semi-natural habitats (e.g. production forest, extensive pastures) are recognized as the stronghold for the majority of species on Earth [7, 8], although some species can use human modified ecosystems [9, 10]. In fact, many threatened species only occur in natural habitats, making the loss of natural habitat a more severe problem for those species [11]. Consequently, the extent of (semi)natural land cover (NLC) is a key determinant of species conservation status in a given landscape.

However, the different relations between NLC and biodiversity are a contentious issue in the debate surrounding the importance of and need for natural areas to conserve biodiversity. For example, studies on plant richness come to different conclusions, with some showing a linear increase with natural habitat cover [12], others a quadratic relationship with the highest species richness in areas with intermediate cover of natural habitats [13], and yet others a lack of correlation between plant species richness and natural habitat cover [14]. This is presumably because the importance of NLC likely varies between species and possibly groups of species, for example, threatened vs non-threatened [15] or native vs exotic species. A better understanding of the importance of NLC to different species and species groups is urgently needed to improve biodiversity conservation in the Anthropocene.

Natural areas may include different ecosystems and habitat types, from forests to shrublands and open grasslands to wetlands. The abiotic conditions vary substantially between these ecosystems ranging from shady, moist forests to sunlit, warm and dry open habitats, and thus provide very different living conditions for species [16, 17]. While an increased amount of forest cover may result in high plant species richness in general [18], not all species prefer habitats with high forest cover [19]. Richness of some shade-intolerant plant species decreases with forest cover [20], as those species prefer more open habitats. Consequently, to assess the importance of natural habitats for species conservation, it may be important to discriminate between natural areas consisting of natural forest areas (NLC-F) or of natural open areas (NLC-O).

In this study, we assess the importance of NLC to the conservation of plant species in the Netherlands. We aim at evaluating the importance of habitat with different extent of NLC for the occurrence, rather than fitness or richness, of certain species. This method will provide us with novel insight into the occurrence probability of almost all species along the proportion of NLC across the landscapes [21]. We address the following sub-questions: (1) does the extent of NLC associate with native plant species' occurrence? (2) do different types of natural areas (forests versus open habitats) affect the conservation status (presence) of native plants? (3) given the importance of natural areas to conservation-relevant species, we ask whether there is a correlation between the status of the plant species (native/exotic, threatened/not threatened and rare/common) and the preferred amount of natural area (or forest or open) in the landscape? (4) Does more NLC make landscapes suitable to more species of conservation-relevance?

## Method

### Study area

Our study area is the Netherlands, for which long-term and detailed monitoring data of biodiversity exist. Most of the country consists of human-dominated landscapes with agriculture

(about 62%) and urban and industrial areas (about 15%), while few (semi-)natural landscapes remain (about 23%).

## Land cover

Compared to non-natural habitats, which can also be potential habitats for plants, NLC in this paper includes all natural or semi-natural habitats (S1 Table) receiving little human disturbance and management. To derive a complete national dataset from which NLC could be obtained, we combined data from three national sources on land use and land cover: *Informatiemodel natuurbeheer* [22], *basisregistratie gewaspercelen* [23], and *bestand bodemgebruik* [24] using ESRI ArcGIS Desktop 10.2 (https://desktop.arcgis.com/en/). These three datasets included 175 land cover classes at the country scale. Marshall [25] aggregated these 175 classes into 16 land cover classes at a resolution of 10x10 m. We further re-classified these 16 classes into '(semi-)natural land cover' and 'non-natural land cover' (S1 Table). Since this study focuses on the terrestrial ecosystems and wetland ecosystems, we excluded the land cover class of 'open water' (large bodies of surface waters, whether fresh or brackish).

Next, we calculated the proportion of NLC within each 1 x 1 km cell in the Netherlands. The NLC-F and proportion of NLC types except for natural forest, further referred to as the NLC-O, were also calculated in order to compare the preferences of plant species for natural forest and natural open habitats (S1 Table, Fig 1A–1C). Based on calculated proportions of NLC, NLC-F and NLC-O in each grid, the proportion ($F_i$) of grids in the Netherlands with different NLC (or NLC-F or NLC-O) equalling $i$ was calculated by

$$F_i = N_i / N \qquad \text{Eq1}$$

where $N_i$ is the number of grids with NLC (or NLC-F or NLC-O) equalling $i$ and $N$ is the total number of grids. To avoid artefacts relating to the species-area relationship from affecting our results, only cells where open water was <10% (equalling a land surface proportion of = >90% in each cell) were included in our analysis (see S1 Text for a sensitivity analysis, S1 Fig).

## Plant species data

We used presence-only records for vascular plants, which include seed plants, conifers, ferns and clubmosses, in the Netherlands from the Dutch National Database of Flora and Fauna (NDFF, www.ndff.nl) collected between the period 2010–2017. Our analysis used species-level taxonomy, so we excluded observations at genus and family level, as well as nomina dubia (e.g. '*Geranium dissectum* / *molle* / *pusillum*'). Species with an 'extinct' or 'disappeared' status [26] were also excluded. Included observations were either point observations or polygons with areas smaller than 3 ha, and the latter indicated the presence of a species within a given area. All polygons were converted to points by taking the centroid of the polygon. Next, each observation was attributed to a 1 x 1 km cell. This left us with 4 773 313 observations (S2 Fig), including 1 128 native and 416 exotic species (S2 Table). All of these species are accepted in the Netherlands, and their corresponding names in the World Flora Online were also included in S2 Table for comparison. Exotic species, which include many cultivated species, were included in this study to compare with native species. Although many exotic species are cultivated, they have been a real part of the environment after introduction to the Netherlands. Finally, duplicate species records were removed to leave a single presence value for each species in each 1 x 1 km cell in the dataset.

## Species distribution modeling

To gain insight in the response of plant species to NLC, NLC-F and NLC-O, we developed species distribution models (SDMs), which have been validated to be an effective method to link

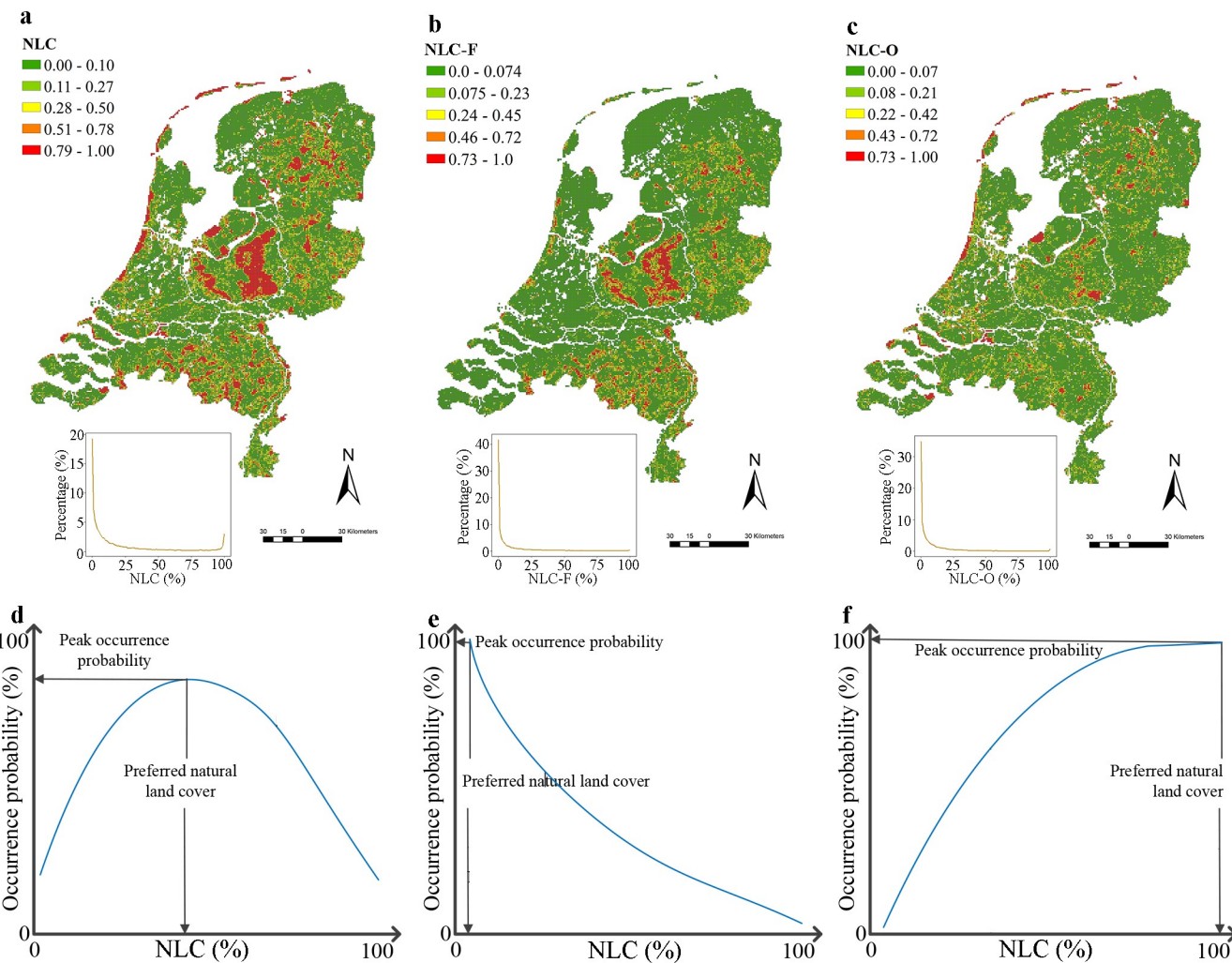

**Fig 1. Land cover map and illustration of plant species responding to NLC.** a, the distribution of NLC across the Netherlands. b, the distribution of the NLC-F across the Netherlands. c, the distribution of the NLC-O across the Netherlands. d-f, Illustration of three types of hypothetical curves indicating plant species responding to NLC (or NLC-F or NLC-O).

environmental variables to species distributions [21]. By using SDMs, we assessed the species occurrence probability rather than fitness or richness and found out which extent of NLC is the best habitat for certain species. In addition to the three variables of interest (NLC, NLC-F and NLC-O), we also included bioclimatic data and soil types. We extracted 19 bioclimatic variables (bio1-bio19) from the WorldClim 2.1 database with resolution 30 seconds [27]. Due to multi-collinearity between these bioclimatic variables, we used principal components analysis (PCA) to convert 19 bioclimatic variables into 5 linearly uncorrelated variables [28], which capture 93% of the variability in the full suite of 19 bioclimatic variables (S3 Table). A national layer of soil types (10 classes produced in 2006) was obtained from https://www.wur.nl/nl/show/Grondsoortenkaart.htm. This was used to generate a rasterized coverage of 10 soil types at 1 km resolution using ESRI ArcGIS Desktop 10.6.1 (https://desktop.arcgis.com/en/). Due to strong correlations between NLC and NLC-F, as well as NLC and NLC-O (S4 Table), we included either NLC or NLC-F and NLC-O in two separate SDMs (i.e. SDM1 includes NLC, five bioclimatic variables and ten soil types; SDM2 includes NLC-F, NLC-O, five bioclimatic variables and ten soil types).

We employed maximum entropy modelling [29] (MaxEnt), which can robustly deal with a variety of presence-only data [29–31] and has outperformed most other SDM modelling applications in dealing with small samples and preventing overfitting [32, 33]. MaxEnt models were developed using the function *maxent* from the R package dismo [34] in R version 3.5.1 [35]. For each species, two separate SDMs were run as mentioned above. For each variable included in the model, we calculated the response of species occurrence probability to this variable, which will be used in further analysis. We used the permutation importance for evaluating the importance of different variables (NLC, etc.) [36]. The validation of the robustness of species distribution prediction was carried out with AUC generated by SDMs [37]. However, AUC values indicating SDM accuracy based on presence-background data are flawed. Hence, we tested the validity of MaxEnt models for each species by comparing the AUC value to a null model [38]. To generate the null model for a species, we constructed 100 MaxEnt models with the same parameterization and structure of the corresponding empirical model, but now using an equal number of randomly drawn occurrence records [39]. We then computed the 95% C. I. AUC value based on yielded 100 AUC values from each null model and assessed whether AUC values of empirical MaxEnt models were higher than their corresponding 95% C.I. AUC values of the fitted null models [38]. Those species for which the AUC values of SDMs were not significantly higher than null models were excluded from further analyses due to the species occurrence not significantly explained by the predictors included in the SDM.

## Response of species to NLC and preferred NLC

Of each species for which the occurrence was significantly explained by the SDM, the shape of the response to NLC (or NLCF or NLCO) was determined, by classifying the curve into 'Increase' (positive linear model), 'Decrease' (negative linear model), 'Unimodal' (quadratic model), 'U-shaped' and 'None' (a response curve with no relationship with either NLC, NLC-F or NLC-O). To get the shape of the response of each species, we assessed the corresponding NLC (or NLC-F or NLC-O) values to the peak and lowest occurrence probability values, using the following criteria: (i) we assigned an increasing response curve if the NLC (or NLC-F or NLC-O) value at peak occurrence probability value equalled 100% and the NLC value at the lowest occurrence probability value equalled 0%; (ii) we assigned a decreasing response curve if the NLC value at peak occurrence probability value equalled 0% and the NLC value at the lowest occurrence probability value equalled 100%; (iii) we assigned a u-shaped response curve if the NLC value at the lowest occurrence probability value ranged between 1% and 99%; (iv) we assigned a unimodal response curve if the NLC value at peak occurrence probability value ranged between 1% and 99%; (v) we assigned a flat response curve ('None') if the NLC value at peak occurrence probability value equalled the NLC value at the lowest occurrence probability value. Response curves of 1 525 species with either decreasing, unimodal or increasing shapes were presented in S3 Fig. 10. No species performed u-shaped response curves and species with a neutral (without a relationship, named 'None') response were omitted as not preferring any NLC. Secondly, for each species with a decreasing, increasing or unimodal relationship with natural cover, we determined their preference for nature (or natural forest or open). This 'preferred natural land cover' (preferred NLC, NLC-F or NLC-O) represents the percentage of NLC (or, NLC-F or NLC-O) at the predicted peak occurrence probability (Fig 1D–1F). For those species with decreasing or increasing relations with NLC (or NLC-F or NLC-O), their preferred NLCs (preferred NLC-Fs or NLC-Os) are always '0%' or '100%', respectively. For species with unimodal shapes, their preferred NLCs (preferred NLC-Fs or NLC-Os) are the NLCs (NLC-Fs or NLC-Os) corresponding to the peak occurrence probability of the predicted response curve. Plant species with unimodal response curves

differed substantially in preferred NLC (and in preferred NLC-Fs or NLC-Os). For illustrative purposes and to facilitate comparisons between plant responses, we divide them in four classes: (i) unimodal0-25%, with peak of unimodal curve at (0%, 25%) NLC (or NLC-F or NLC-O); (ii) unimodal25-50%, with peak at [25%, 50%) NLC; (iii) unimodal50-75%, with peak at [50%, 75%) NLC; and (iv) unimodal75-100%, with peak at [75%, 100%) NLC. In the comparisons, we distinguished six groups (Increasing curve, decreasing curve and the four unimodal groups) based on preferred NLC, NLC-F or NLC-O.

## Species occurrence in landscape cells (1x1km) relative to the distribution of NLC in those cells nationwide

To assess whether the occurrence of species followed the distribution of NLC in the Netherlands or not, the proportion of observations of species within group $k$ (one of six groups described above) occurring in grid cells with NLC equalling $i$ ($F_{ik}$) was calculated by

$$F_{ik} = N_{ik}/N_k. \qquad \text{Eq2}$$

where $N_{ik}$ is the number of observations of species within group $k$ and occurring in grids with NLC equalling $i$. $N_k$ is the number of all observations of species within group $k$.

Finally, the representativeness of the occurrence of a species group along NLC, describes the difference between the proportion of occurrences of grid cells with different NLC in the Netherlands and the relative occurrence of each species group along NLC. The difference in representativeness ($R_{ik}$), was calculated by

$$R_{ik} = F_{ik} - F_i. \qquad \text{Eq3}$$

## Association between species category and preferred NLC, NLC-F and NLC-O

To assess whether different species groups differ in their response to natural cover, we classified all species based on their (1) origin (native and exotic); (2) threatened status (threatened and not threatened); (3) rarity (rare and common); (4) growth form (woody and herbaceous). Data on threatened status, rarity and species origin were obtained from the Red List of Vascular Plants of the Netherlands [26]. Ninety-three percent of the data about growth form were extracted from online existing databases [40–42] and 7% were manually assigned using images on Google Image (https://images.google.com/) and Wikipedia (https://www.wikipedia.org/). Since we were interested in differences between main categories, for the threatened status of species, we omitted Data Deficient (DD), and reclassified Least Concern (LC) and Near Threatened (NT) as 'not threatened' and Vulnerable (VU), Endangered (EN) and Critically Endangered (CR) as 'threatened'.

We then compared how these three broad species categories are related to four groups of preferred NLC, NLC-F and NLC-O (0–25%, 25–50%, 50–75% and 75–100%). We used Contingency Analyses (Pearson's) to assess differences between levels within each category. All analyses and visualisations were conducted in R version 3.5.1 [35].

## Average occurrence probability increase

The increase in occurrence probability per plant group was calculated by

$$OP_i = \sum_{j=1}^{j=n}(OP_{ij} - OP_{0j})/n. \qquad \text{Eq4}$$

where $OP_i$ is the averaged occurrence probability increase by increasing NLC from 0% to i%, $OP_{ij}$ is the occurrence probability of species j at NLC i%, $OP_{0j}$ is the occurrence probability of

species j at NLC 0%. n is the total number of species in a certain group (e.g. threatened species).

## Results

### NLC, NLC-F and NLC-O patterns across the Netherlands

To explore the relationships between NLC and species' presence, we first quantified the NLC, NLC-F and NLC-O across space and mapped their frequency distributions (Fig 1A–1C). Both NLC-F and NLC-O showed a sharply decreasing pattern. Although NLC showed a 'U-shaped' pattern, the proportion of grid cells with 100% NLC was still much lower than that with 0% NLC. Taken together, the frequency distribution of NLC, NLC-F and NLC-O at the 1x1 km scale revealed that 90% of the landscapes in the Netherlands have less than 73% NLC, less than 39% NLC-F and less than 29% NLC-O.

### NLC correlates to plant species occurrence

We analyzed the relationships between native plant species occurrence and NLC. Out of 1 128 native plant species, SDMs for 1 122 species were significantly better than the null model and those species showed either a decreasing, unimodal or increasing relationship with NLC (Fig 2A). Results of all 1 122 species are in S2 Table and response curves of these species are in S3 Fig. Based on permutation importance value, NLC was, on average, a more important variable for explaining species occurrence than other variables (Table 1). Of the 1 122 native plant species, 20 times more species showed a linear increase with NLC (21.4%, 240 spp.) than a linear decrease (1%, 11 spp.) with NLC (Fig 2A, 2B and 2D). The majority of species (77.6%, 871 spp.) showed a unimodal response curve (Fig 2A and 2C) with NLC.

Moreover, the steepness of the response curves varied widely between different species (Fig 2), even between species with the same response curve shape (either increase, unimodal or decrease). For example, both *Carex extensa* and *Carex distans* had unimodal relationships with NLC, but the curve of *Carex extensa* was much steeper than that of the *Carex distans* (S3 Fig).

We further calculated the preferred NLC, i.e. the NLC corresponding to the peak occurrence probability in the response curve, to portray where these native plant species are most likely to occur. The median and mean of preferred NLC of the 1 122 species were 61.0% and 65.6%, respectively (Fig 2I). In total, 840 of 1 122 species (accounting for 74.9) preferred NLC above 50%, with 346 of those preferring NLC above 75% (Fig 2J). What's more, threatened and rare species particularly preferred landscapes with high NLC. Of the threatened species, 193 of 230 (accounting for 83.9%) preferred NLC higher than 50%, with 120 of those species preferring NLC higher than 75% (Fig 3, S4 Fig). Of the rare species, 388 of 503 (accounting for 77.1%) preferred NLC higher than 50%, with 233 of those species preferring NLC higher than 75% (Fig 3). However, when assessing the proportion of grid cells with different NLC in the Netherlands (Fig 1, the yellow line in 2, and 3), grid cells with NLC above 50% only account for 15.6% of all grid cells in the Netherlands, while these concurred with the preferred NLC of 74.9% native species, 83.9% threatened species and 77.1% rare species. In short, there is a strong mismatch between the NLC available in the Netherlands and the preference for landscapes with higher NLC of the majority of native plant species (Fig 3).

Since most species showed unimodal shapes, but their preferences for NLC varied widely (ranging from 10.4% to 96.1%, Fig 2K), we divided them into four groups based on NLC preference of 0–25%, 25–50%, 50–75% and 75–100% (Fig 2E–2H). The percentages of species in the four groups were 1.3%, 22.6%, 44.0% and 9.4% respectively (the sum is not 100% as species with decreasing and increasing response curves were excluded) (Fig 2J). Overall, the frequency distribution of preferred natural cover of species with unimodal responses followed a normal

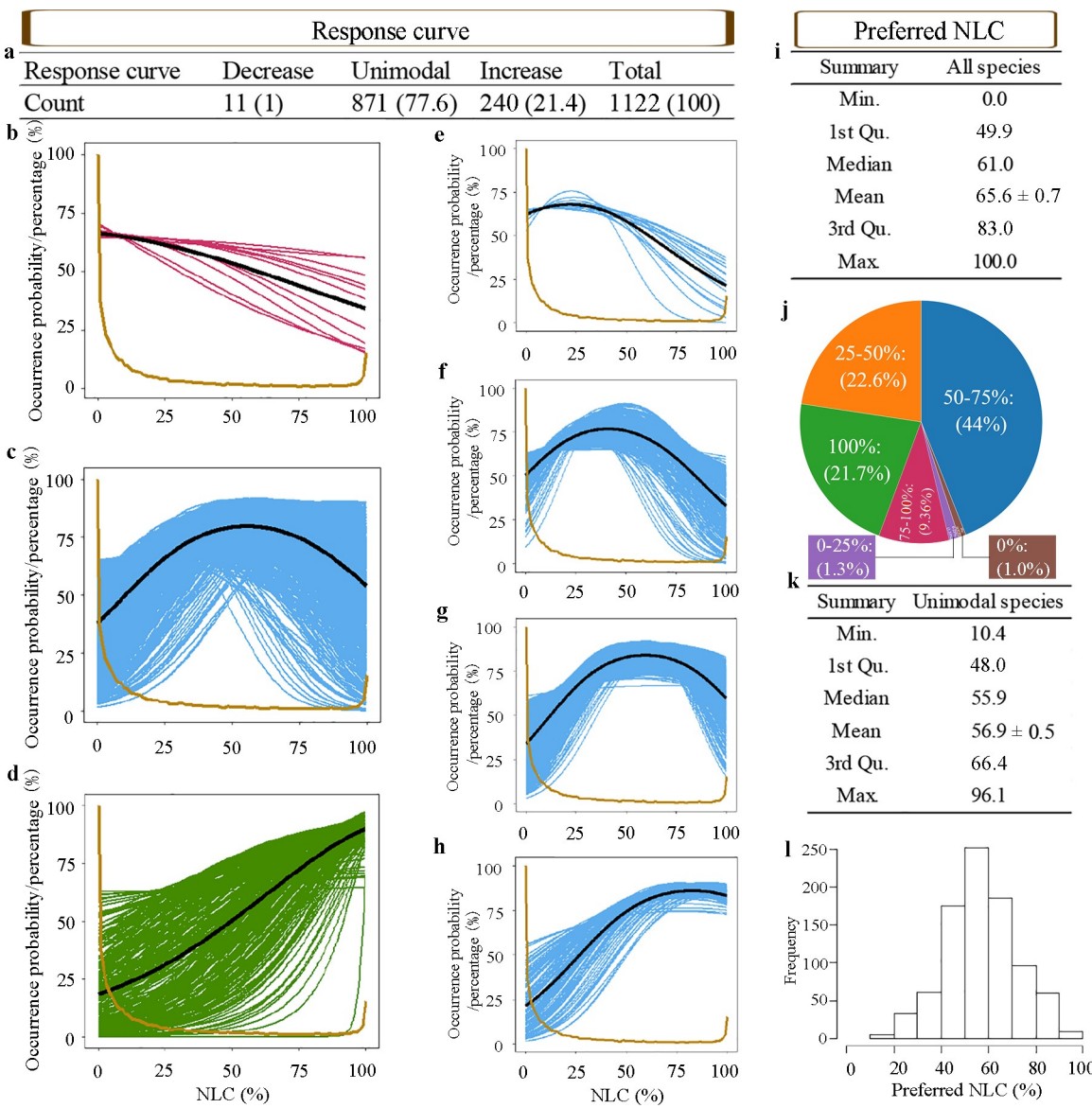

**Fig 2. Response to NLC for 1 122 native plant species with significant null models.** 3 species with a neutral (without a relationship, named 'None') response were omitted as not preferring any NLC. a-h, Response curves of plant species to NLC. i-l, Summary statistics of preferred NLCs. a, Summary of all 1 122 native plants responding to NLC (with percentages in parentheses). b-d, Response curves of species with decreasing (each magenta line indicates one species) (b), unimodal (each blue line indicates one species) (c) and increasing (each green line indicates one species) (d) relations with NLC. The dark black line is the average response curve of each species group. The yellow line indicates the standardized proportion of grids with different NLC in the Netherlands. Both the occurrence probability and the standardized proportion (percentage) range from 0% to 100% and are indicated by the y axis. e-h, Species with unimodal shapes were split into four groups based on their preferred NLCs (e, species with preferred NLCs ranging from 0–25%; f, species with preferred NLCs ranging from 25–50%; g, species with preferred NLCs ranging from 50–75%; h, species with preferred NLCs ranging from 75–100%). i-j, Statistics of preferred NLCs of all 1 122 native plant species (i, summary of 1 122 plant species' preferred NLCs; j, percentage of species (in brackets) in different preferred NLC range groups, i.e. 0%, 0–25%, 25–50%, 50–75%, 75–100% and 100%). k-l, Statistics of preferred NLCs of species with unimodal shapes (k, summary of plant species' preferred NLCs; l, histogram of preferred NLCs of plant species with unimodal relations with NLC). All mean values are means ± SE.

distribution, with a median of 55.9% (Fig 2K and 2L). Over 69.1% of these species were more likely to occur in landscapes with NLC more than 50%.

Next, we compared the relative occurrence of each species group (increase, decrease, unimodal0-25%, unimodal25-50%, unimodal50-75% and unimodal75-100%) to the presence

**Table 1. Comparison of permutation importance among variables.**

| Model1 (with NLC) | | All spp. | Native spp. | Exotic spp. |
|---|---|---|---|---|
| | Summary | Mean | Mean | Mean |
| | NLC | 19.58 | 22.79 | 10.90 |
| | bioclim1 | 10.34 | 11.20 | 8.01 |
| | bioclim2 | 13.18 | 14.31 | 10.14 |
| | bioclim3 | 3.37 | 2.72 | 5.11 |
| | bioclim4 | 3.69 | 3.36 | 4.60 |
| | bioclim5 | 2.47 | 2.39 | 2.71 |
| | built_upon_area | 8.76 | 7.25 | 12.86 |
| | heavy_clay | 2.90 | 2.44 | 4.15 |
| | heavy_sabulous_clay | 2.97 | 2.42 | 4.44 |
| | light_clay | 5.93 | 5.34 | 7.54 |
| | light_sabulous_clay | 2.64 | 1.95 | 4.53 |
| | loam | 1.46 | 1.24 | 2.07 |
| | peat | 5.72 | 5.25 | 6.99 |
| | sand | 14.43 | 15.43 | 11.71 |
| | swampy_on_sand | 1.96 | 1.47 | 3.27 |
| | water | 0.58 | 0.44 | 0.96 |
| Model2 (with NLCF+NLCO) | | All spp. | Native spp. | Exotic spp. |
| | Summary | Mean | Mean | Mean |
| | NLC-F | 7.42 | 7.61 | 6.89 |
| | NLC-O | 12.80 | 15.01 | 6.75 |
| | bioclim1 | 10.06 | 10.92 | 7.71 |
| | bioclim2 | 13.44 | 14.53 | 10.47 |
| | bioclim3 | 3.58 | 2.97 | 5.27 |
| | bioclim4 | 3.63 | 3.32 | 4.47 |
| | bioclim5 | 2.47 | 2.34 | 2.80 |
| | built_upon_area | 8.83 | 7.59 | 12.22 |
| | heavy_clay | 2.96 | 2.56 | 4.04 |
| | heavy_sabulous_clay | 3.05 | 2.61 | 4.25 |
| | light_clay | 5.82 | 5.34 | 7.14 |
| | light_sabulous_clay | 2.80 | 2.10 | 4.70 |
| | loam | 1.48 | 1.25 | 2.11 |
| | peat | 5.73 | 5.32 | 6.84 |
| | sand | 13.58 | 14.72 | 10.46 |
| | swampy_on_sand | 1.88 | 1.42 | 3.14 |
| | water | 0.48 | 0.39 | 0.74 |

Higher values indicate a higher importance and thus more explanatory power in the models.

distribution of NLC in the landscapes of the Netherlands. In this way, we can show the extent to which different species groups overuse or underuse grid cells with different NLC. Species groups with increasing response curves and unimodal curves with preferred NLC 75–100% were under-represented in areas with low NLC but over-represented in areas with high NLC (Fig 4). The species group with decreasing response curves and preferred NLC 0–25% were a bit over-represented in areas with low NLC and under-represented in areas with high NLC. Finally, the group with preferred NLC 25–50% was under-represented in low NLC and high NLC but over-represented in moderate NLC, while the group with preferred NLC 50–75% was

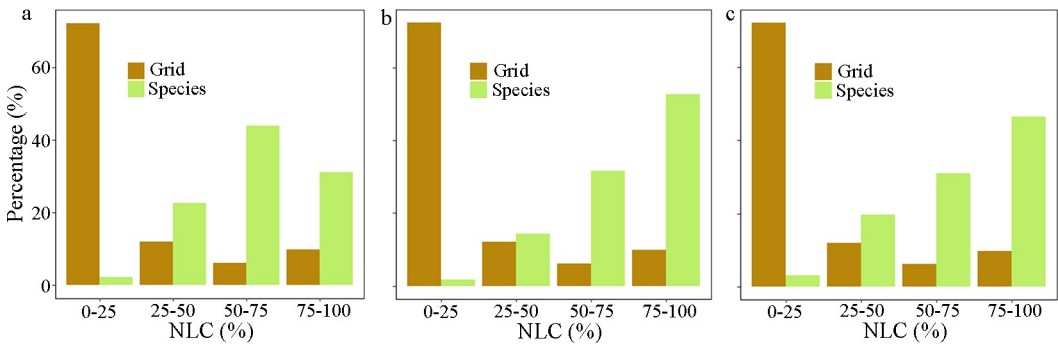

**Fig 3. Percentage of grids with different NLC and species with different preferred NLC in the Netherlands.** a, 1 122 native species. b, 230 threatened species. c, 503 rare species. The brown color means the distribution of grid cells with different NLC and the green color means the distribution of species with different preferred NLC.

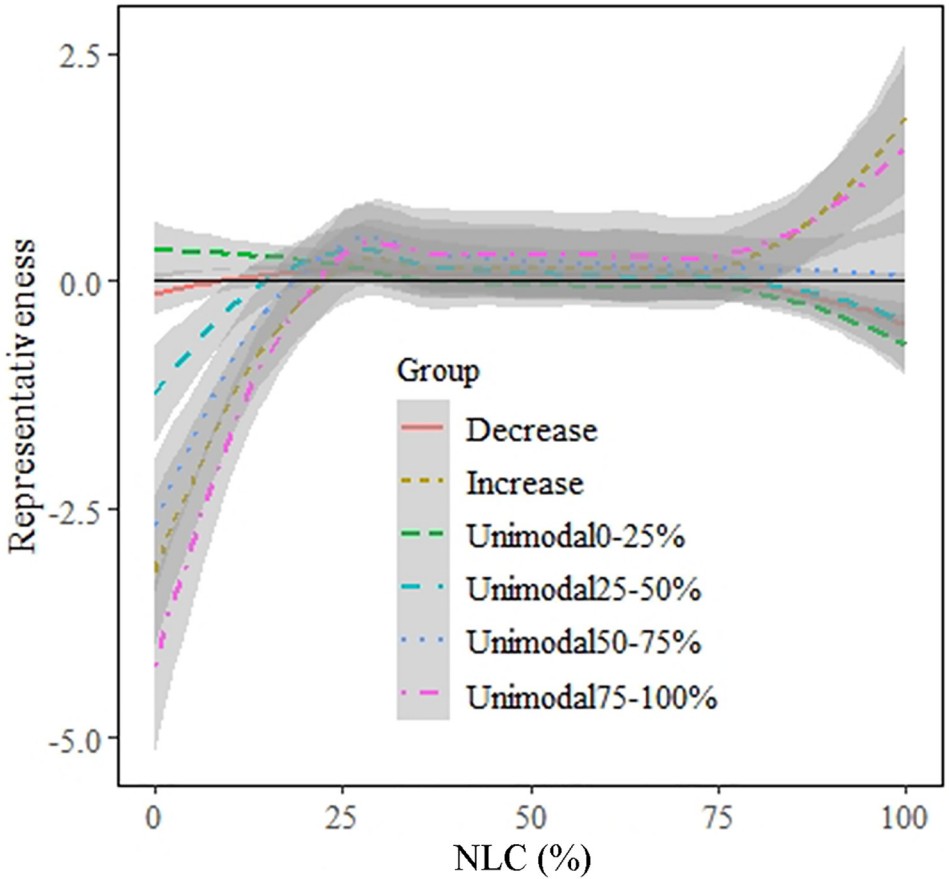

**Fig 4. The representativeness, i.e. relative occurrence of a species group in relation to the NLC availability in the Netherlands.** It describes the difference between the proportion of occurrence of grid cells with different NLC in the Netherlands and the relative occurrence of each species group along NLC. Species were classified into decrease, increase, preferred NLCs of 0–25% (unimodal0-25%), 25–50% (unimodal25-50%), 50–75% (unimodal50-75%) and 75–100% (unimodal75-100%), respectively. Lines represent the best-fit regressions and the grey bands represent the regression 95% confidence interval. The black horizontal line indicated 0.

only under-represented in areas with low NLC. In general, most species were over-represented in areas with high NLC but under-represented in low NLC cells.

## Differences in preferences of plant species for NLC-F or NLC-O

For 1 091 and 1 121 species, of 1 122 native species, the SDM showed meaningful response curves (i.e. SDM models were better than null models) with NLC-F and NLC-O, respectively (S5 and S6 Figs, S2 Table). Average permutation importance showed that these two variables ranked first and fifth respectively (Table 1). According to the response curves of species to NLC-F and NLC-O, we obtained the shapes and preferred NLC-F and NLC-O for each species. We found that most species (76.3%) showed a unimodal response to NLC-F. The mean and median of preferred NLC-F of 1 091 species were 35.9% and 39.9% (S5 Fig); Most species (88.8%) also showed a unimodal response to NLC-O. The mean and median of preferred NLC-O of 1 121 species were 57.6% and 55.0% (S6 Fig).

We compared the preferences for NLC-F and NLC-O of each native species and found that most species preferred areas with 25–75% NLC-F and NLC-O (Fig 5). Threatened and rare species also preferred moderate NLC-F and NLC-O (S7 and S8 Figs). In other words, a minority of native species prefers landscapes with a very high (>75%) NLC-F (47spp., 4.3%) or NLC-O (184spp., 16.9%). Only 9 species (0.8%) showed increasing occurrence probabilities

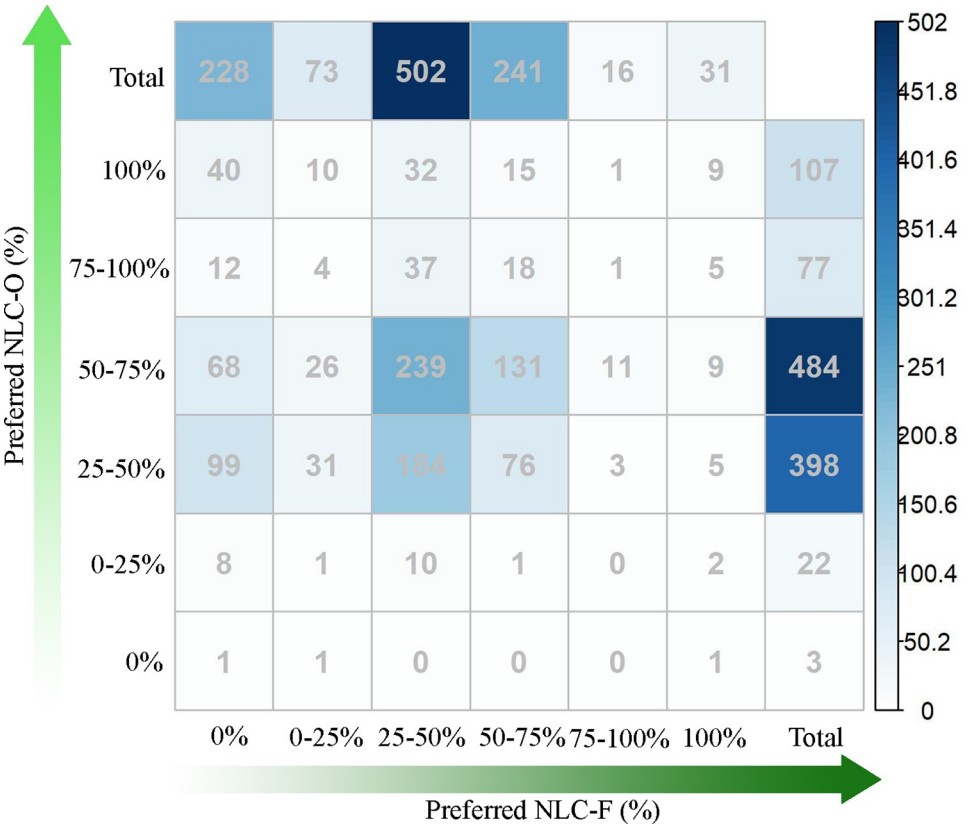

**Fig 5. Preferred NLC-F and preferred NLC-O of native plant species.** Preferred NLC-F and NLC-O were classified into six groups, corresponding to species with decreasing shapes (preferred NLC-F or NLC-O equalling 0%), species with increasing shapes (preferred NLC-F or NLC-O equalling 100%), species with preferred NLC-F or NLC-O between 0% and 25%, species with preferred NLC-F or NLC-O ranging from 25% to 50%, species with preferred NLC-F or NLC-O ranging from 50% to 75%, species with preferred NLC-F or NLC-O ranging from 75% to 100%. The number of species (in each grid) increases with the blue color in each grid getting dark.

with both NLC-F and NLC-O increase. In this case, these species prefer high NLC, but natural types don't matter for these species. One species (0.1%) showed decreasing occurrence probabilities with both NLC-F and NLC-O increase. Forty species (3.7%) showed increasing relationships with NLC-O but decreased relationships with NLC-F. These species prefer natural open areas but not natural forest areas. In contrast, only one species (0.1%) preferred natural forest areas but not natural open areas.

## Importance of nature to plant species varies between threatened and unthreatened as well as between native and exotic species

Finally, we explored whether the preference for NLC, NLC-F and NLC-O varies between species categories. Contingency Analysis revealed that threatened and rare species were more likely to occur in landscapes with NLC, NLC-F and NLC-O more than 75% compared to not threatened and common species (Fig 6A–6F, S9 Fig). Rare species were also more likely to occur in landscapes with NLC-O between 0% and 25% compared to common species.

Most exotic plant species in the Netherlands showed unimodal relationships with NLC, with a peak in the response curve between 25%-50% NLC (S10 Fig). Contingency Analysis indicated that exotic species were more likely to occur in landscapes with NLC and NLC-O lower than 50% compared to native species (Fig 6G–6I, S9 Fig). In contrast, native species were more likely to occur in landscapes with a relatively high NLC and NLC-O (50–75%). Exotic species also preferred landscapes with 0–25% and above 75% NLC-F, but native species preferred landscapes with 25–50% NLC.

## Discussion

### The importance of NLC to plant species' presence

Almost all native plant species in the Netherlands had an association with NLC and more species preferred landscapes with high NLC than low NLC. This nationwide study confirms the importance of NLC to plant species, and it supports the need of protecting natural habitats for species conservation [43]. Of the 1 122 native species, 74.9% (840 species) preferred NLC above 50%. This strongly contrasts with the 15.6% of Dutch landscapes that have NLC higher than 50%. It suggests that most species in the Netherlands lack sufficient suitable landscapes, and this is particularly true for threatened species. The overrepresentation of most species in high NLC landscapes and their underrepresentation in low NLC landscapes further exemplifies the imbalance between current landscapes and species' needs in the Netherlands. It also indicates that any increase in NLC in low NLC landscapes might mitigate species decline and promote species occurrence. This is consistent with the marginal occurrence probability increase, which is high in areas with low NLC (S11 Fig). Although a correlative study such as ours cannot definitely assign causality, the findings from controlled experimental studies at small scale suggest NLC alters plant biodiversity and increase NLC can increase species [12, 13].

Although most species preferred NLC above 50%, more species preferred NLC 50–75% than 75–100%. There were also more species preferring NLC 25–50% than 0–25%. This broad hump-shape trend is consistent with the intermediate disturbance hypothesis [44], which stipulates that natural areas with intermediate levels of land cover change may provide more heterogeneous habitats (higher habitat diversity) harbouring more species than completely natural areas, e.g. continuous deciduous forests, salt marshes. This is also found by another study at large scale, which shows that unimodal relationships exist between Canadian avian species and NLC [45]. However, this relationship is mainly explained by the amount of NLC

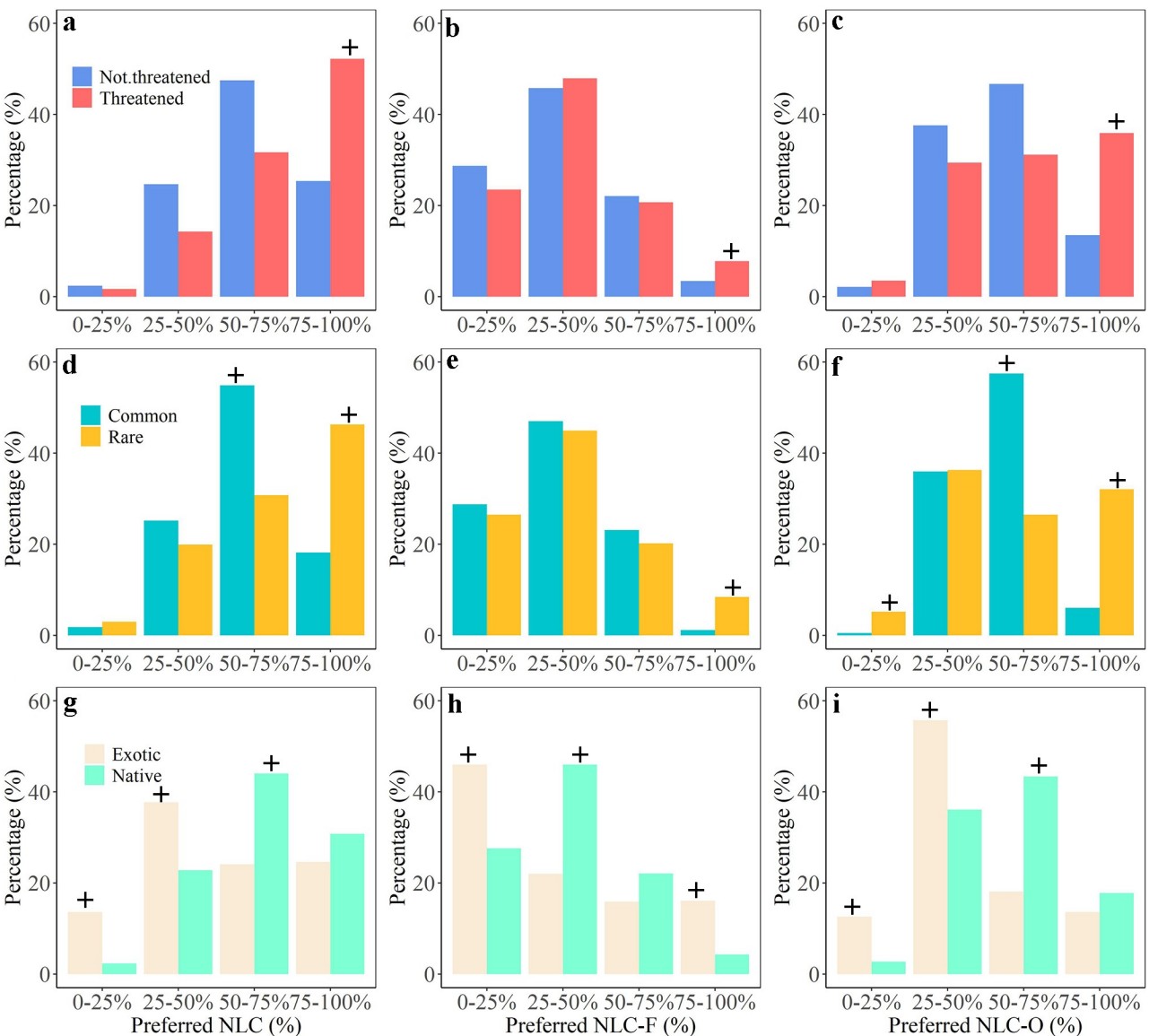

**Fig 6. Percentage distribution of species along the preferred NLC, preferred NLC-F and preferred NLC-O within each species category (threatened status, rarity and origin of species).** a-c, Percentage distribution of threatened and not threatened species along preferred NLC (a), NLC-F (b) and NLC-O (c). d-f, Percentage distribution of common and rare species along preferred NLC (d), NLC-F (e) and NLC-O (f). h-i, Percentage distribution of native and exotic species along preferred NLC (g), NLC-F (h) and NLC-O (i). '+' indicates values that are above expectation according to our Contingency Analysis (see supplementary analysis for detailed results).

per se and not by increased habitat diversity. These patterns are in line with the 'habitat amount hypothesis' which states that species respond primarily to the total habitat amount at the landscape level and that there is little additional effect of the habitat configuration on species [46]. Threatened plant species were even more prevalent in landscapes with 75–100% NLC than other native plants, which concurs with the findings of Berg et al [15]. These findings reveal that landscapes with very high NLC are particularly valuable for the protection of threatened species.

Unsurprisingly, individual plant species showed very different relations with NLC and the most common response was a preference for landscapes with 50–75% NLC. Following on

from this, we assume that species richness is also the highest in landscapes with 50–75% NLC. Indeed, the relationship of species richness with NLC based on observations was quadratic, which is consistent with the predicted richness-NLC relationships based on species-level responses (Supplementary Method, S12 Fig). Although our study does not give any insight into whether and how individual species affect the response of community species richness to NLC, some findings about the traits of individual species and their assemblage dictating the responding of the community to the environment (e.g., climate) have been observed before [47].

Given the relatively long red lists of threatened species/rare species and the sparsity of natural habitats in the Netherlands, governmental and private conservation initiatives to strengthen the conservation of plant species should strive to increase the cover of natural habitats and the presence of natural elements in degraded landscapes and sustain currently protected areas. This is because landscapes with NLC always harbour more species than landscapes without NLC (Fig 7). However, more NLC in a landscape will not automatically result in more native species because there is often a non-linear increase in species occurrence probability when going from 0% to 100% NLC. The optimum NLC lies between 64–100%, being generally higher for threatened species (with optimum at 80% NLC) and rare species (with optimum at 70% NLC), particularly very rare species (with optimum at 100% NLC), than for not threatened and common species (Fig 7B–7D). Exotic species show a different pattern with the highest occurrence in landscapes with lower NLC than that of native species (Fig 7A). Thus, increasing NLC in totally disturbed landscapes (0% NLC) to 64% or even more is likely to increase species richness overall, but different species groups require different restoration strategies.

### Effect of open or forested natural landscapes on species conservation

Natural habitats and landscapes range from open grasslands and wetlands, through mixed landscapes to closed forests. Since abiotic conditions vary widely between these landscapes [16, 17], we analysed the preference of plant species for more open natural areas (higher NLC-O) or forested natural areas (higher NLC-F). As expected, native plant species have different preferences for one or the other (e.g. *Alisma gramineum* preferred 0% NLC-F but 65% NLC-O). However, most plant species preferred moderately open or closed natural areas rather than areas with completely open or forested nature. Our findings are consistent with ecological theory [44, 48, 49], as well as empirical results of light preference indicated by, for example Ellenberg's Indicator [50], where most species have moderate light preference values. This result means that a mixture of forest and open cover suits more species than a high forest cover or completely open cover. Clearly, from an individual species' perspective, conclusions may be different, but our study disproves the often heard argument that creating more forests as part of the climate adaptation agenda is beneficial to biodiversity conservation at large, which has also been criticized by the scientific community elsewhere [51]. Similarly, it disproves the statement that biodiversity at large needs agriculture as the vast majority of the Dutch plant species prefers landscapes with substantial, i.e. >50%, natural habitats. This fits with the recent opinion that the EU should integrate mosaics of diverse ecosystems to realize the biodiversity strategy rather than massive tree planting [52]. However, compared to totally disturbed landscapes (0% NLC-F or NLC-O), plants occurring in forested natural landscapes show a slightly different relationship than plants occurring in open natural landscapes (Figs 8 and 9). Landscapes with NLC-O show similar occurrence patterns as NLC, namely they always harbour more native species than landscapes without NLC-O (Fig 8). However, landscapes with 100% forest harbour fewer native species than completely disturbed landscapes (Fig 9).

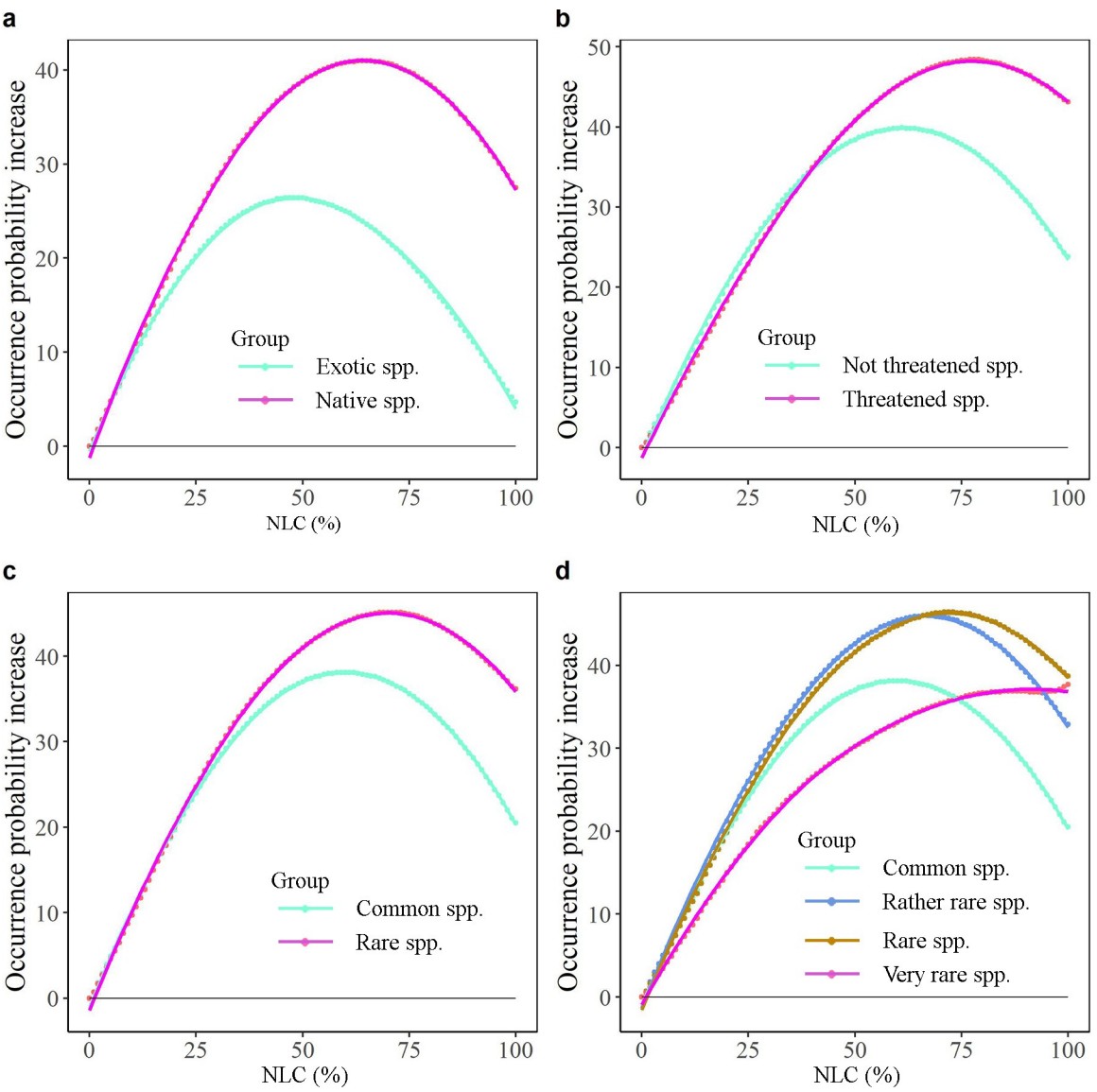

**Fig 7. The average occurrence probability change (average occurrence probability change from 0% NLC to others, e.g. NLC at 1x1 km resolution increases from 0% to 11%) of different species groups.** a, All native species vs exotic species. b, Threatened species vs not threatened species. c, Rare species vs common species. d, Rare species vs common species, but rare species are classified into three categories (i.e. very rare, rare, rather rare) according to the Red List of Vascular Plants of the Netherlands [26]. Lines represent the best-fit regressions.

Landscapes with 100% forest or open harbour fewer exotic species than completely disturbed landscapes.

## Preference of threatened, rare and exotic species for natural landscapes

Much of the effort in nature conservation is aimed at rare or threatened species [53], e.g. those on the EU Natura 2000 lists. Our results support this investment, as threatened and rare species are more dependent on natural landscapes than other species. This is probably because threatened and rare species have more specific abiotic preferences or are more susceptible to

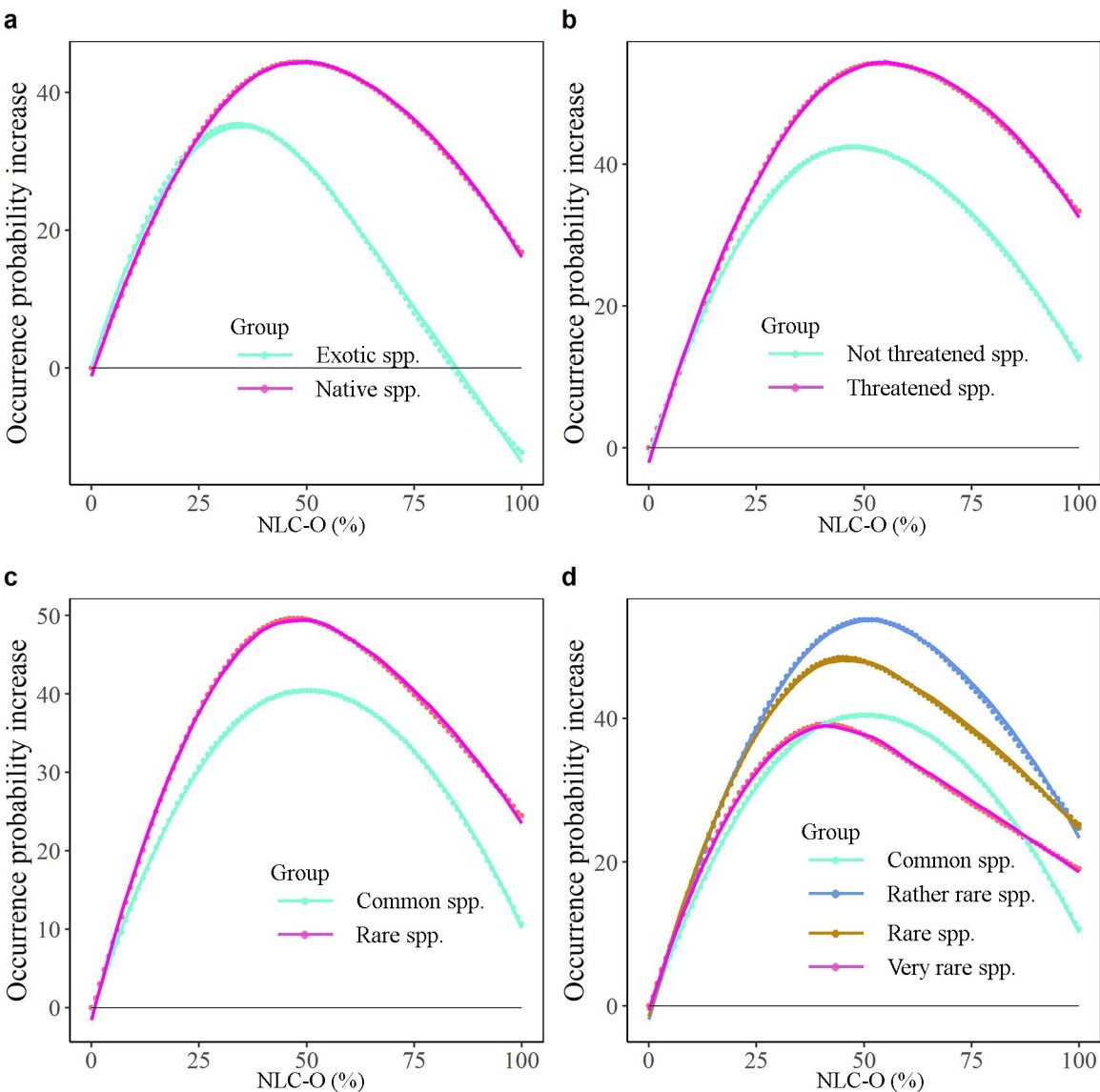

**Fig 8. The average occurrence probability change (average occurrence probability change from 0% NLC-O to others, e.g. the NLC-O at 1x1 km resolution increases from 0% to 11%) of different species groups.** a, all native species vs exotic species. b, threatened species vs not threatened species. c, rare species vs common species. d, Rare species vs common species, but rare species are classified into three categories (i.e. very rare, rare, rather rare) according to the Red List of Vascular Plants of the Netherlands [26]. Lines represent the best-fit regressions.

disturbances and interspecific competition [53, 54], and conditions may be met more often in high NLC areas (e.g. protected areas [55]). However, some natural open renimants in human dominated areas can also be refuge of some rare species [56, 57].

Exotic species are less likely to occur in landscapes with relatively high NLC and NLC-O. One possible reason is that they are more capable of invading disturbed habitats [58] but less so in high NLC areas with more native species [59–63], where interspecific competition may be stronger [64]. Exotic woody species also had a higher occurrence probability in highly forested areas. This may be an artifact from the overrepresentation of woody species in the Dutch exotic flora (S13 Fig).

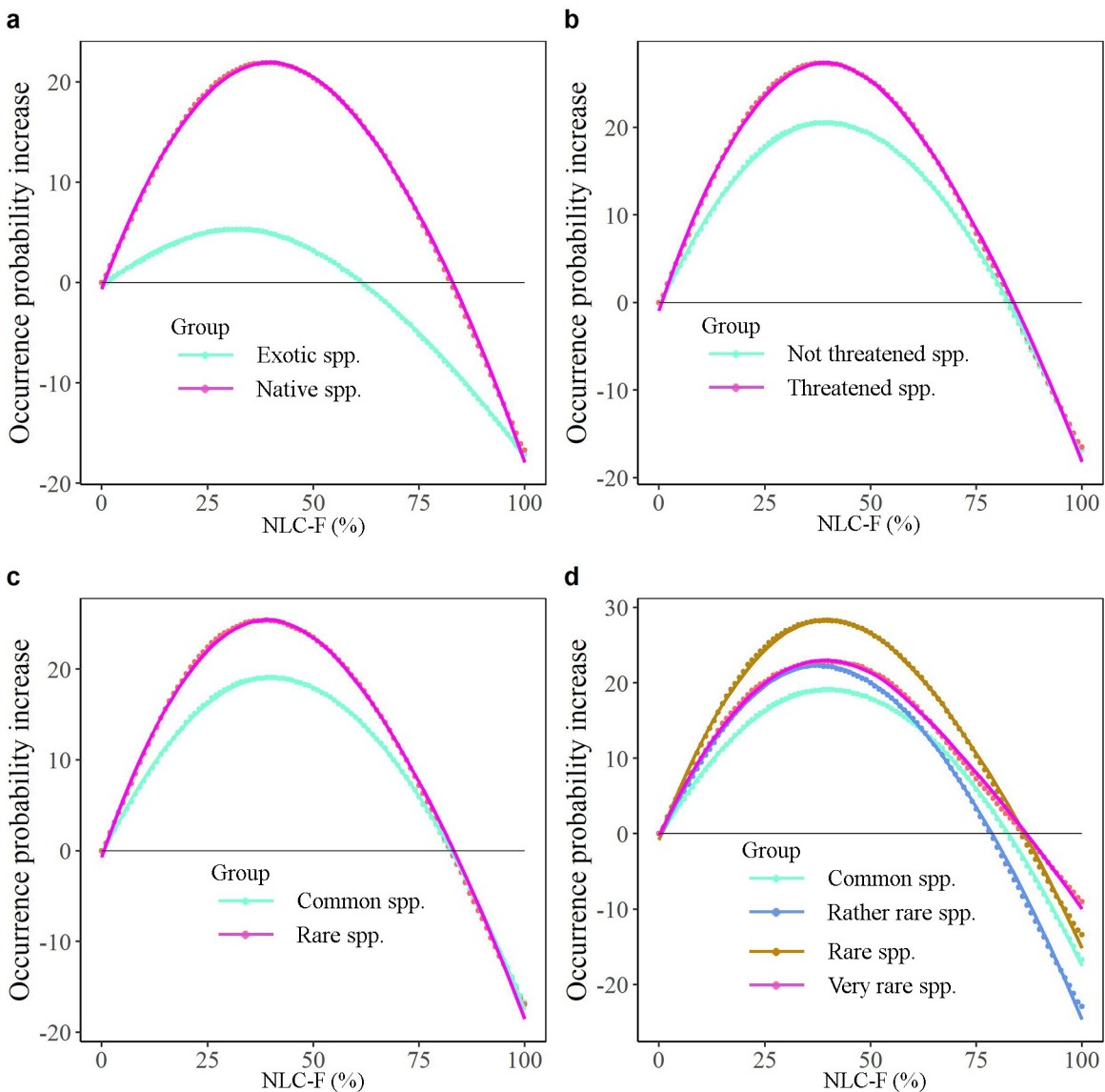

**Fig 9. The average occurrence probability change (average occurrence probability change from 0% NLC-F to others, e.g. the NLC-F increases from 0% to 11%) of different species groups.** a, all native species vs exotic species. b, threatened species vs not threatened species. c, rare species vs common species. d, Rare species vs common species, but rare species are classified into three categories (i.e. very rare, rare, rather rare) according to the Red List of Vascular Plants of the Netherlands [26]. Lines represent the best-fit regressions.

## Comparing the effect of NLC to soil types and bioclimate

Although we focus on species conservation by exploring the relationships between land cover and plant species, we also compared the effect of land cover factors to other factors (ten soil types and five bioclimatic variables), the importances of which have been indicated by many studies (e.g. [65]). Our results indicate that NLC is often the most important factor to plant species presence, although the importance decreases when we consider NLC-F and NLC-O separately. One reason for NLC to be so important, may be its close relation to local microclimate [66–68], in combination with the relatively similar macroclimatic conditions across the Netherlands. In general, NLC modulates macroclimatic conditions (i.e. WorldClim-derived data) and thus affects plant species presence [16, 69] in addition to climate. Interestingly, NLC

was much more important than soil and bioclimatic factors for the presence of native species, while the opposite was true for exotic species (Table 1).

## Conclusion and management implications

In conclusion, this nationwide study assessing an unprecedented number of species has shown that NLC is strongly related to plant species' presence. Interestingly, particularly in the light of ongoing loss of NLC, this study is the first to quantify these relationships. Most species, particularly species of conservation concern, prefer landscapes with higher than 50% NLC (e.g. natural wetland, grassland, heathland or forest). This is a strong contrast to the current landscapes in the Netherlands, i.e. few landscapes have NLC higher than 50%. Thus, protecting natural landscapes, particularly landscapes with substantial NLC, is greatly beneficial to species conservation and should be the priority goal. On this basis, increasing the area of natural elements in landscapes will increase the occurrence probability of most species, with the highest marginal increase in strongly disturbed landscapes. Moreover, most species prefer natural areas with a mixture of forest and open areas. Thus, a landscape with a mixture of different natural types is better for plant biodiversity in general, and also for threatened species, than an area with one type of natural cover (natural forest or open area). Following mixed landscapes, areas with natural open cover should have the higher priority of conservation than with natural forest cover since threatened and rare species are more likely to occur in landscapes with high NLC-O but low NLC-F. Finally, conservation initiatives should aim to increase NLC, by the combination of maintaining current natural cover and restoring disturbed landscapes. While landscapes with approx 60% will conserve common native species, high levels (>80%) will best aid very rare and threatened species. We hope that this work will help government and conservation agencies to improve conservation guidelines, focus their efforts on natural areas and increasing natural elements and natural habitat types in artificial areas and in this way stand a better chance to realize their national and regional biodiversity strategy targets. Our method can also be used in other regions to help local governments identify species of particular concern from the stressor of natural areas decrease and identify landscapes with which extent of NLC does species prefer.

## Supporting information

**S1 Text.**
(DOCX)

**S1 Table. Land cover types used in this study.**
(DOCX)

**S2 Table. Full table of results.**
(XLSX)

**S3 Table. PCA for 19 bioclimatic variables.**
(DOCX)

**S4 Table. Spearman's test for all 18 variables included in this research.**
(XLSX)

**S1 Fig. The average occurrence frequency based on grid cells with more than 90% land cover vs grid cells with 100% land cover.** Each point indicates one plant species. The x axis means the average occurrence frequency based on all grid cells with 100% land cover. The y axis means the average occurrence frequency based on all grid cells with more than 90% land

cover.
(DOCX)

**S2 Fig. 4 773 313 observations of plant species.**
(DOCX)

**S3 Fig. Response curves of 1 525 plant species (both native and exotic species) to natural land cover.**
(PDF)

**S4 Fig. 230 threatened plant species responding to natural land cover (NLC).** 2 species with a significantly neutral (without a relationship, named 'None') response were omitted as not preferring any NLC. **a-h**, Response curves of plant species to NLC. **k-l**, Summary statistics of preferred NLCs. **a**, Summary of all 230 threatened plants responding to NLC (with percentages in parentheses). **b-d**, Response curves of species with increasing (each red line indicates one species) (**b**), unimodal (each blue line indicates one species) (**c**) and decreasing (each green line indicates one species) (**d**) relations with NLC. The dark black line is the average response curve of each species group. The yellow line indicates the standardized proportion of grids with different NLCs in the Netherlands. Both the occurrence probability and the standardized proportion (percentage) range from 0% to 100% and are indicated by the y axis. **e-h**, Species with unimodal shapes are split into four categories based on their preferred NLCs (**e**, species with preferred NLCs ranging from 0–25%; **f**, species with preferred NLCs ranging from 25–50%; **g**, species with preferred NLCs ranging from 50–75%; **h**, species with preferred NLCs ranging from 75–100%). **i-j**, Statistics of preferred NLCs of all 230 threatened plant species (**i**, summary of 230 plant species' preferred NLCs; **j**, percentage of species in different categories, including 0%, 0–25%, 25–50%, 50–75%, 75–100% and 100%). **k-l**, Statistics of preferred NLCs of species with unimodal shapes (**k**, summary of plant species' preferred NLCs; **l**, histogram of preferred NLCs of plant species with unimodal relations with NLC). All mean values are means ± SE.
(DOCX)

**S5 Fig. 1 091 native plant species responding to the cover of natural forest area (NLC-F).** 24 species with 'U-shaped' responses were omitted as not ecologically realistic, and 6 species with a significantly neutral (without a relationship, named 'None') response were omitted as not preferring any NLC-F. **a-h**, Response curves of plant species to NLC-F. **k-l**, Summary statistics of preferred NLC-Fs. **a**, Summary of all 1 091 native plants responding to NLC-F (with percentages in parentheses). **b-d**, Response curves of species with decreasing (each red line indicates one species) (**b**), unimodal (each blue line indicates one species) (**c**) and increasing (each green line indicates one species) (**d**) relations with NLC-F. The dark black line is the average response curve of each species group. The yellow line indicates the standardized proportion of grids with different NLC-Fs in the Netherlands. Both the occurrence probability and the standardized proportion (percentage) range from 0% to 100% and are indicated by the y axis. **e-h**, Species with unimodal shapes are split into four categories based on their preferred NLC-Fs (**e**, species with preferred NLC-Fs ranging from 0–25%; **f**, species with preferred NLC-Fs ranging from 25–50%; **g**, species with preferred NLC-Fs ranging from 50–75%; **h**, species with preferred NLC-Fs ranging from 75–100%). **i-j**, Statistics of preferred NLC-Fs of all 1 091 native plant species (**i**, summary of 1 091 plant species' preferred NLC-Fs; **j**, percentage of species in different categories, including 0%, 0–25%, 25–50%, 50–75%, 75–100% and 100%). **k-l**, Statistics of preferred NLC-Fs of species with unimodal shapes (**k**, summary of plant species' preferred NLC-Fs; **l**, histogram of preferred NLC-Fs of plant species with unimodal

relations with NLC-Fs). All mean values are means ± SE.
(DOCX)

**S6 Fig. 1 121 native plant species responding to the cover of natural open area (NLC-O).** 4 species with 'U-shaped' responses were omitted as not ecologically realistic. **a-h,** Response curves of plant species to NLC-O. **k-l,** Summary statistics of preferred NLC-Os. **a,** Summary of all 1 121 native plants responding to NLC-O (with percentages in parentheses). **b-d,** Response curves of species with decreasing (each red line indicates one species) (**b**), unimodal (each blue line indicates one species) (**c**) and increasing (each green line indicates one species) (**d**) relations with NLC-O. The dark black line is the average response curve of each species group. The yellow line indicates the standardized proportion of grids with different NLC-O in the Netherlands. Both the occurrence probability and the standardized proportion (percentage) range from 0% to 100% and are indicated by the y axis. **e-h,** Species with unimodal shapes are split into four categories based on their preferred NLC-Os (**e**, species with preferred NLC-Os ranging from 0–25%; **f**, species with preferred NLC-Os ranging from 25–50%; **g**, species with preferred NLC-Os ranging from 50–75%; **h**, species with preferred NLC-Os ranging from 75–100%). **i-j,** Statistics of preferred NLC-Os of all 1 121 native plant species (**i**, summary of 1 121 plant species' preferred NLC-Os; **j**, percentage of species in different categories, including 0%, 0–25%, 25–50%, 50–75%, 75–100% and 100%). **k-l,** Statistics of preferred NLC-Os of species with unimodal shapes (**k**, summary of plant species' preferred NLC-Os; **l**, histogram of preferred NLC-Os of plant species with unimodal relations with NLC-Os). All mean values are means ± SE.
(DOCX)

**S7 Fig. Preferred NLC-F and NLC-O of native plant species, which are classified into threatened and not threatened species. a**, Two-dimensional plot indicating the preferred NLC-F and NLC-O of each species. Each dot indicates one species. Red dots indicate threatened species and blue dots indicate not threatened species. **b**, Number of dots within each subgroup along the x axis (0%, 0–25%, 25–50%, 50–75%, 75–100% and 100%). **c**, Number of dots within each subgroup along the y axis (0%, 0–25%, 25–50%, 50–75%, 75–100% and 100%).
(DOCX)

**S8 Fig. Preferred NLC-F and NLC-O of native plant species, which are classified into rare and common species. a**, Two-dimensional plot indicating the preferred NLC-F and NLC-O of each species. Each dot indicates one species. Yellow dots indicate rare species and turquoise dots indicate common species. **b**, Number of dots within each subgroup along the x axis (0%, 0–25%, 25–50%, 50–75%, 75–100% and 100%). **c**, Number of dots within each subgroup along the y axis (0%, 0–25%, 25–50%, 50–75%, 75–100% and 100%).
(DOCX)

**S9 Fig. Contingency analysis on whether threatened status and origin affect the preferred NLC, NLC-F and NLC-O. a1-a3**, Differences of preferred NLC within groups of threatened status, rarity and origin. **b1-b3**, Differences of preferred NLC-F within groups of threatened status, rarity and origin. **c1-c3**, Differences of preferred NLC-O within groups of threatened status, rarity and origin. Preferred NLC, NLC-F and NLC-O mean the preferences for natural land cover (preferred NLC), the cover of natural forest area (preferred NLC-F) and the cover for natural open area (preferred NLC-O).
(DOCX)

**S10 Fig. 403 exotic plant species responding to natural land cover (NLC).** 7 species with a significantly neutral (without a relationship, named 'None') response were omitted as not

preferring any NLC. **a-h**, Response curves of plant species to NLC. **k-l**, Summary statistics of preferred NLCs. **a**, Summary of all 403 exotic plants responding to NLC (with percentages in parentheses). **b-d**, Response curves of species with decreasing (each red line indicates one species), unimodal (each blue line indicates one species) and increasing (each green line indicates one species) relations with NLC. The dark black line is the average response curve of each species group. The yellow line indicates the standardized proportion of grids with different NLC in the Netherlands. Both the occurrence probability and the standardized proportion (percentage) range from 0% to 100% are indicated by the y axis. **e-h**, species with unimodal shapes are split into four categories based on their preferred NLCs (**e**, species with preferred NLCs ranging from 0–25%; **f**, species with preferred NLCs ranging from 25–50%; **g**, species with preferred NLCs ranging from 50–75%; **h**, species with preferred NLCs ranging from 75–100%). **i-j**, Statistics of preferred NLCs of all 403 exotic plant species (**i**, summary of 403 plant species' preferred NLCs; **j**, percentage of species in different categories, including 0%, 0–25%, 25–50%, 50–75%, 75–100% and 100%). **k-l**, Statistics of preferred NLCs of species with unimodal shapes (**k**, summary of plant species' preferred NLCs; **l**, histogram of preferred NLCs of plant species with unimodal relations with NLC). All mean values are means ± SE.
(DOCX)

**S11 Fig. The average marginal occurrence probability change of different species groups.** a, All native species vs exotic species. b, Threatened species vs not threatened species. c, Rare species vs common species. d, Rare species vs common species, but rare species are classified into three categories (i.e. very rare, rare, rather rare) according to the Red List of Vascular Plants of the Netherlands [1]. Lines represent the best-fit regressions. The average marginal occurrence probability change means the average occurrence probability change with 1% natural land cover increase, e.g. natural land cover at 1x1 km resolution increases from 10% to 11%).
(DOCX)

**S12 Fig. Plants responding to natural land cover (NLC) at species-level and community-level. a**, The occurrence probability of 1 122 native plant species responding to NLC. Each grey line indicates one species, and the dark line means the average response of all 1 122 native species. **b**, Predicted relationships of richness responding to NLC, with a best fitting quadratic model. **c**, Relationships of plant species richness responding to NLC based on observations, with a best fitting quadratic model. Lines represent the best-fit regressions.
(DOCX)

**S13 Fig. The percentage of woody and herbaceous species within each species category (native species vs exotic species).**
(DOCX)

## Acknowledgments

We specifically thank Maarten van 't Zelfde for providing help with spatial analysis in land cover and information about soil types.

## Author Contributions

**Conceptualization:** Kaixuan Pan, Ellen Cieraad, Geert R. de Snoo, Koos Biesmeijer.

**Data curation:** Kaixuan Pan, Koos Biesmeijer.

**Formal analysis:** Kaixuan Pan, Merijn Moens, Leon Marshall, Ellen Cieraad, Geert R. de Snoo, Koos Biesmeijer.

**Funding acquisition:** Kaixuan Pan.

**Methodology:** Kaixuan Pan, Merijn Moens, Leon Marshall, Ellen Cieraad, Geert R. de Snoo, Koos Biesmeijer.

**Project administration:** Kaixuan Pan, Geert R. de Snoo, Koos Biesmeijer.

**Resources:** Kaixuan Pan, Leon Marshall, Koos Biesmeijer.

**Software:** Kaixuan Pan, Merijn Moens, Leon Marshall.

**Supervision:** Ellen Cieraad, Geert R. de Snoo, Koos Biesmeijer.

**Visualization:** Kaixuan Pan.

**Writing – original draft:** Kaixuan Pan, Ellen Cieraad, Koos Biesmeijer.

**Writing – review & editing:** Kaixuan Pan, Merijn Moens, Leon Marshall, Ellen Cieraad, Geert R. de Snoo, Koos Biesmeijer.

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
