## [Decision Letter · Decision Letter 0]

10 May 2021

PONE-D-21-08410

Importance of natural land cover for plant species’ conservation: a nationwide study in The Netherlands

PLOS ONE

Dear Dr. Pan,

Thank you for submitting your manuscript to PLOS ONE. After careful consideration, we feel that it has merit but does not fully meet PLOS ONE’s publication criteria as it currently stands. Therefore, we invite you to submit a revised version of the manuscript that addresses the points raised during the review process.

We look forward to receiving your revised manuscript.

Kind regards,

Daniel de Paiva Silva, Ph.D.

Academic Editor

PLOS ONE

Additional Editor Comments:

Dear Pan et al.

After three independent reviews, I believe you manuscript may be published in PLoS One should the issues raised by the three reviewers are taken care of in the next version of your MS. All three reviewers raised important issues to improve your MS, especially reviewer #1. Review all of your data, your analyses, and your supplmentary materials.

Considering the pandemic, please resubmit a new version of your text, along with a detailed rebuttal letter informing to the reviewers the performed changes by August 10 2021. If more time is needed, please let me know. If you are able to resubmit before the established due date, do not hesitate to do it so.

Sincerely,

Daniel Silva

Journal Requirements:

3. We note that Figure 1 in your submission contain map images which may be copyrighted. All PLOS content is published under the Creative Commons Attribution License (CC BY 4.0), which means that the manuscript, images, and Supporting Information files will be freely available online, and any third party is permitted to access, download, copy, distribute, and use these materials in any way, even commercially, with proper attribution. For these reasons, we cannot publish previously copyrighted maps or satellite images created using proprietary data, such as Google software (Google Maps, Street View, and Earth). For more information, see our copyright guidelines: http://journals.plos.org/plosone/s/licenses-and-copyright.

You may seek permission from the original copyright holder of Figure 1 to publish the content specifically under the CC BY 4.0 license. 

If you are unable to obtain permission from the original copyright holder to publish these figures under the CC BY 4.0 license or if the copyright holder’s requirements are incompatible with the CC BY 4.0 license, please either i) remove the figure or ii) supply a replacement figure that complies with the CC BY 4.0 license. Please check copyright information on all replacement figures and update the figure caption with source information. If applicable, please specify in the figure caption text when a figure is similar but not identical to the original image and is therefore for illustrative purposes only.

Reviewers' comments:

Reviewer's Responses to Questions

**Comments to the Author**

1. Is the manuscript technically sound, and do the data support the conclusions?

Reviewer #1: Partly

Reviewer #2: Yes

Reviewer #3: Yes

2. Has the statistical analysis been performed appropriately and rigorously? 

Reviewer #1: I Don't Know

Reviewer #2: Yes

Reviewer #3: Yes

3. Have the authors made all data underlying the findings in their manuscript fully available?

Reviewer #1: Yes

Reviewer #2: Yes

Reviewer #3: Yes

4. Is the manuscript presented in an intelligible fashion and written in standard English?

Reviewer #1: Yes

Reviewer #2: Yes

Reviewer #3: Yes

5. Review Comments to the Author

Reviewer #1: In this manuscript Pan et al use biodiversity data, species distribution modelling and natural land cover data to explore the relationship between natural land cover and species occurrence diversity for spermatophyte plants in the Netherlands.

At line 23 the acronym ‘NLC’ is used without first being defined. Whilst the acronym is defined later-on in the introduction (lines 42-43), I wonder if the acronym could either be defined in the abstract OR simply not used in the abstract?

With these methods can they really attempt to test sub-question (4) “how much NLC is necessary to protect conservation-relevant species?” -- surely to some extent this depends on the exact species of plant. Different species are not exactly comparable units. A large tree species will definitely need more NLC area than an endangered buttercup (Ranunculus) that does not strictly require NLC habitat -- these can grow in urban environments e.g. as roadside vegetation. There can be a lot of roadside verge habitat within 1x1km grid cells that are categorized as not natural land cover. Until additional factors are taken into consideration e.g. how large each plant is and how it lives, you will not get a good unstanding of the role of NLC (as just one factor among many).

Taxonomic precision

An examination of supplementary table 2 demonstrates that _only_ the binomial name is given for each species. There is no taxonomic authority supplied with each of these binomial names. In most cases this won’t matter but there are a few where it might be ambiguous. Best practice would be to provide the binomial name AND the taxonomic authority for each to be as precise as possible e.g. Ocimum canescens A.J. Paton

I took the liberty of running checks against all 1563 binomial names provided in supp. table 2 with the help of Taxonstand, matching to The Plant List v1.1 At least eleven of the binomial names given have more than one valid synonym, and thus greater taxonomic precision (via the provision of authority detail) is necessary to resolve what exact species concept the authors intend to use. Those eleven are: Juncus ambiguus, Potamogeton mucronatus, Arenaria leptoclados, Festuca arenaria, Orobanche picridis, Bryonia dioica, Carex oederi, Carex ovalis, Rubus fruticosus, Tilia x vulgaris, and Centaurea montana. Full .csv output from these basic data checks are available as a gist on github here: https://gist.github.com/rossmounce/e5a4c5b2b4642496ee716cf3fb9d2e20

The species scope of the study is also inadequately described - there are no bryophytes or mosses or fern species included in this study thus it is not a study covering ‘land plants’ (Embryophytes) but instead it covers terrestrial seed plants (spermatophytes) inclusive of some gymnosperms e.g. pine species but it is mostly composed of angiosperm species. They should also be more explicit that aquatic plant species such as duckweed (e.g. Wolffia) are strangely included in this study. Surely all the strictly-aquatic plant species should be removed from this analysis if the central theses to be tested are about the type of _land cover_ ? Water is not land.

Why does the manuscript talk about 1563 plant species (line 122) whilst the supplementary materials explain that supplementary figure 2 is “Response curves of 1544 plant species”. Please explain or correct this discrepancy.

Taxonomic name cleaning

Taxonomic name cleaning was apparently absent from this study. Which is a big flaw that should be corrected before publication. Many of the species binomials included in this study are not valid names according to The Plant List v1.1 . In total 108 of the 1563 are synonyms according to TPL v1.1 . For instance “Ranunculus ficaria” is either Ficaria verna Huds. according to TPL v1.1 or Caltha palustris L. according to Plants of the World Online (POWO). Please clean all taxonomic names to either TPL v1.1 or POWO or the Leipzig Catalogue of Vascular Plants (LCVP) and reanalyse the data. This is very important - some of those species aren’t recognized as full species anymore, it undermines the thesis of some of these analyses to not be using _only_ taxonomically accepted/robust species.

Neither POWO nor TPL v1.1 recognize “Rosa henkeri-schulzei” as an accepted species.

Similarly Salix ‘Sekka’ is not taxonomically valid or precise binomial name.

Line 213, states that data was obtained on (1) threatened status, (2) rarity, and (3) species origin from the Red List of Vascular Plants of Netherlands (2012). I can see that threatened status (column U) and species origin (column T) data are given in supplementary table 2 but where is the rarity data they used - it does not seem to be contained within supplementary table 2? Can the authors please make sure that the rarity assertion data they used is provided within supplementary table 2.

I’m surprised to see no information given about higher level taxonomy for each of the 1563 species in supplementary table 2, nor any real discussion of it in the manuscript. Surely it would be interesting to know which are gymnosperms, which are angiosperms, are there patterns to be observed within and between species at the family level (or lower). Surely ‘grass’ (Poaceae) species might have a different set of results relative to ‘pine’ (Pinaceae) species?

Although this journal does not strictly require the authors to provide the code underlying the analyses presented, I think it would certainly put this reviewer more at ease if some of underlying code used in this analysis was provided, so that we could better see what has been done computationally on this data. Please consider sharing some or all of the underlying code.

I hereby knowingly waive my right to anonymity and intentionally sign my peer review report,

Ross Mounce

Reviewer #2: The manuscript analyzed the importance of natural land cover to the occurrence of plant species in the Netherlands through species distribution models. The paper is well organized. The methods and analyses in the manuscript are robust. This is a potentially useful case study, but it is of uneven quality at present. Below are some of my suggestions.

1. I think if Fig. 1a was a map of plant species richness, it could provide more information. I’d also like to see pictures of species richness of different plant groups. Besides, could the authors please demonstrate grid cells with different proportions of natural land cover with different colors in Fig. 1b?

2. There are 1,563 species in Supplementary Table 2 while 1,544 species in Supplementary Fig. 2. Please clarify why the numbers don't tally.

3. In line 257-259, the authors wrote “b-d, Response curves 258 of species with increasing (each magenta line indicates one species) (b), unimodal (each blue line indicates one 259 species) (c) and decreasing (each green line indicates one species) (d) relations with NLC”. However, I think Fig. 2b showed the decreasing and Fig. 2d showed the increasing relationship with increasing NLC.

4. Please add a horizontal line to signify the line of “Representiveness = 0.0” in Fig. 3.

5. Most contents of the discussion were about the preferences of species of different groups for natural landscapes, but the relevant figures were placed mostly in Supporting Information. Meanwhile the authors classified species with unimodal response curves into four classes and showed several figures in the result section though with little discussion later. I think it’s better to reduce this part of the result charts and exchange with some figures regarding to preferences of different species groups.

6. In line 351-352, the authors wrote “Contingency Analysis revealed that threatened and rare species were more likely to occur in 352 landscapes with NLC, NLC-F and NLC-O more than 75%”. However, Fig. 5b and 5e showed that threatened and rare species were more likely to occur in landscapes with NLC-F less than 75%. The conclusions drawn from Fig. 5 were not convincing and rigorous.

7. The authors must have their supporting information carefully edited. There are some errors such as “rariy” in Supplementary Figure 9 and several whitespace problems.

8. I suggest the authors put Supplementary Table 5 in the manuscript, not in the Supplementary information.

9. Discussion 4.1 and 4.2 seems a little repetitive, the authors need to reorganize these two parts.

Reviewer #3: Based on the detailed and comprehensive data, this study carried out a comprehensive study on the relationship between natural land cover types and species distribution in the Netherlands. The topics concerned by the authors have certain guiding significance for the formulation of plant diversity protection strategies and decision-making at the regional scale. However, I think the authors combined different subjects in the article, which may greatly weaken the value and significance of this study. I suggest that the author divide this research at least two papers. One is to discuss the distribution status and laws of natural land cover types from different classification perspectives. The second is the relationship between different types of gradient coverage and the distribution of species diversity of different groups. At present, because the author combines the two topics together, it is difficult to get the Internal connection between them. It is suggested that the authors should revise the paper. Other minor issues are listed below:

1. Line 23: The full name of NLC should be provided. Otherwise, readers don’t know what NLC is if they don't read the text.

2. Line 30: The word ‘informs’ should be ‘inform’.

3. Lines 81-82: the words ‘natural land cover (NLC)’ should be ‘NLC’. The authors should standardize the use of abbreviations. For the words NLC, NLC –F and NLC-O, authors always used their abbreviations, or their full names, even both of them in through article. This is not in line with writing the general norms of writing.

6. PLOS authors have the option to publish the peer review history of their article (what does this mean?). If published, this will include your full peer review and any attached files.

Reviewer #1: **Yes: **Ross Mounce

Reviewer #2: No

Reviewer #3: No

---

## [Author Response · Author response to Decision Letter 0]

28 Jul 2021

Responses to reviewers

Journal Requirements:

 RE: We have revised the paper based on the detailed requirements;

RE: The dataset generated for supporting the analysis in this study is included in the Supplementary Information. The raw data of species occurrence are available from the Dutch National Database of Flora and Fauna (NDFF, www.ndff.nl) but restrictions apply to the availability of these data, which were used under license for the current study, and so are not shared publicly. This restriction has been imposed by the Dutch National Database of Flora and Fauna. However, these data are available upon valid request from NDFF service team, serviceteamNDFF@natuurloket.nl. In fact, Dutch government is currently moving towards making the NDFF data publicly available. 

3. We note that Figure 1 in your submission contain map images which may be copyrighted. All PLOS content is published under the Creative Commons Attribution License (CC BY 4.0), which means that the manuscript, images, and Supporting Information files will be freely available online, and any third party is permitted to access, download, copy, distribute, and use these materials in any way, even commercially, with proper attribution. For these reasons, we cannot publish previously copyrighted maps or satellite images created using proprietary data, such as Google software (Google Maps, Street View, and Earth). For more information, see our copyright guidelines: http://journals.plos.org/plosone/s/licenses-and-copyright.

RE: Figure 1 is made by the authors based on an aggregation of public data (species occurrence data and land-use data). Therefore the copyright is with us.

RE: We have updated our manuscript according to the guidelines.

Reviewer #1: In this manuscript Pan et al use biodiversity data, species distribution modelling and natural land cover data to explore the relationship between natural land cover and species occurrence diversity for spermatophyte plants in the Netherlands.

1. At line 23 the acronym ‘NLC’ is used without first being defined. Whilst the acronym is defined later-on in the introduction (lines 42-43), I wonder if the acronym could either be defined in the abstract OR simply not used in the abstract?

RE: Thank you! We have given the full definition instead of abbreviation in the abstract. 

2. With these methods can they really attempt to test sub-question (4) “how much NLC is necessary to protect conservation-relevant species?” -- surely to some extent this depends on the exact species of plant. Different species are not exactly comparable units. A large tree species will definitely need more NLC area than an endangered buttercup (Ranunculus) that does not strictly require NLC habitat -- these can grow in urban environments e.g. as roadside vegetation. There can be a lot of roadside verge habitat within 1x1km grid cells that are categorized as not natural land cover. Until additional factors are taken into consideration e.g. how large each plant is and how it lives, you will not get a good understanding of the role of NLC (as just one factor among many).

RE: We partly agree with you that a per plant species analysis will provide different answers for each species. We decided to approach this at the level of all plant species in a landscape (Fig 7). 

To clarify this, we have revised this sub-question as “Does more natural land cover (NLC) make landscapes suitable to more species of conservation-relevance?”. 

3. Taxonomic precision

An examination of supplementary table 2 demonstrates that _only_ the binomial name is given for each species. There is no taxonomic authority supplied with each of these binomial names. In most cases this won’t matter but there are a few where it might be ambiguous. Best practice would be to provide the binomial name AND the taxonomic authority for each to be as precise as possible e.g. Ocimum canescens A.J. Paton

I took the liberty of running checks against all 1563 binomial names provided in supp. table 2 with the help of Taxonstand, matching to The Plant List v1.1 At least eleven of the binomial names given have more than one valid synonym, and thus greater taxonomic precision (via the provision of authority detail) is necessary to resolve what exact species concept the authors intend to use. Those eleven are: Juncus ambiguus, Potamogeton mucronatus, Arenaria leptoclados, Festuca arenaria, Orobanche picridis, Bryonia dioica, Carex oederi, Carex ovalis, Rubus fruticosus, Tilia x vulgaris, and Centaurea montana. Full .csv output from these basic data checks are available as a gist on github here: https://gist.github.com/rossmounce/e5a4c5b2b4642496ee716cf3fb9d2e20

RE: The status of taxonomic names has been a continuous source of debate since Linnaeus started naming species. Currently, there are many initiatives that, often in parallel, try to solve the disagreements between different lists and opinions [Co-author Biesmeijer is, as the scientific director of Naturalis Biodiversity Center, one of the largest European taxonomic institutes, working on accepted solutions]. We have chosen to keep the original name list of the Netherlands Database Flora and Fauna, the governmental dataholder. We double checked the list and found 8 cases where 6 names (accepted and synonym) were still in the NDFF database erroneously. We removed the synonym taxa. In other cases the NDFF and the Dutch Flora list recognize 2 taxa (at species level) whereas the world plant list recognizes 1 taxon. In that case we kept the 2 Dutch taxa as separate as they are recorded as separate (and distinguishable) taxa by recorders. For comparison, we have added the reference in the World Flora to the supplemental table 2 so that names can be referred to their entry there (even though the World Flora is not completely resolved either). Analysis has been reconducted.

4. The species scope of the study is also inadequately described - there are no bryophytes or mosses or fern species included in this study thus it is not a study covering ‘land plants’ (Embryophytes) but instead it covers terrestrial seed plants (spermatophytes) inclusive of some gymnosperms e.g. pine species but it is mostly composed of angiosperm species. They should also be more explicit that aquatic plant species such as duckweed (e.g. Wolffia) are strangely included in this study. Surely all the strictly-aquatic plant species should be removed from this analysis if the central theses to be tested are about the type of _land cover_ ? Water is not land.

RE: We do not agree with the referee as we analysed species within the vascular plants (Tracheophyta) including seed plants (spermatophytes), conifers (gymnosperms; Pinus etc), ferns (Polypodiopsida; Dryopteris, Asplenium etc) and clubmosses (Lycopodiopsida; e.g. Lycopodium, Lycopodiella). Mosses are not included as they are not vascular plants. We have indicated this in a more taxonomically correct way (line 112-114). 

The Netherlands consists mostly of water. While we excluded the large water bodies (virtually no vascular plants are present in water of more than 2m depth), most habitats are filled with small streams, ditches, ponds, marsh or wetland areas. All plants occurring in these ‘land areas’ (with small waters) are included in the analysis if they meet the other criteria. These include aquatic vascular plants indeed.

5. Why does the manuscript talk about 1563 plant species (line 122) whilst the supplementary materials explain that supplementary figure 2 is “Response curves of 1544 plant species”. Please explain or correct this discrepancy.

RE: We ran models for 1563 species in total, but for 1554 species we could make significant response curves (i.e. increasing, decreasing and unimodal). For the other species the models were not significantly explaining their distribution. That’s why only responses of 1544 species are finally reported and were used for further analysis. 

6. Taxonomic name cleaning

Taxonomic name cleaning was apparently absent from this study. Which is a big flaw that should be corrected before publication. Many of the species binomials included in this study are not valid names according to The Plant List v1.1 . In total 108 of the 1563 are synonyms according to TPL v1.1 . For instance “Ranunculus ficaria” is either Ficaria verna Huds. according to TPL v1.1 or Caltha palustris L. according to Plants of the World Online (POWO). Please clean all taxonomic names to either TPL v1.1 or POWO or the Leipzig Catalogue of Vascular Plants (LCVP) and reanalyse the data. This is very important - some of those species aren’t recognized as full species anymore, it undermines the thesis of some of these analyses to not be using _only_ taxonomically accepted/robust species.

Neither POWO nor TPL v1.1 recognize “Rosa henkeri-schulzei” as an accepted species.

Similarly Salix ‘Sekka’ is not taxonomically valid or precise binomial name.

RE: Thank you for your suggestion! As what we stated above, we used the plant IDs used by the national database of the Netherlands, NDFF and all of those data have been validated by experts. They are accepted species in the Netherlands. However, we also gave the taxonomic names according to the World Flora in the updated S2 table.

7. Line 213, states that data was obtained on (1) threatened status, (2) rarity, and (3) species origin from the Red List of Vascular Plants of Netherlands (2012). I can see that threatened status (column U) and species origin (column T) data are given in supplementary table 2 but where is the rarity data they used - it does not seem to be contained within supplementary table 2? Can the authors please make sure that the rarity assertion data they used is provided within supplementary table 2.

RE: Two columns were missing in S2 table. I have updated it.

8. I’m surprised to see no information given about higher level taxonomy for each of the 1563 species in supplementary table 2, nor any real discussion of it in the manuscript. Surely it would be interesting to know which are gymnosperms, which are angiosperms, are there patterns to be observed within and between species at the family level (or lower). Surely ‘grass’ (Poaceae) species might have a different set of results relative to ‘pine’ (Pinaceae) species?

RE: In this study we focus on species and their conservation status. In this sense each species is by itself and while it would be interesting to add the taxonomic information, we feel the tables are already too large. In a next paper we plan to assess the link between all kinds of traits including phylogenetic allegiance.

9. Although this journal does not strictly require the authors to provide the code underlying the analyses presented, I think it would certainly put this reviewer more at ease if some of underlying code used in this analysis was provided, so that we could better see what has been done computationally on this data. Please consider sharing some or all of the underlying code.

RE: For the species distribution modelling we used the R package DISMO as we indicated in the text. If readers are interested in the other analyses the code could be made available upon request. 

Reviewer #2: The manuscript analyzed the importance of natural land cover to the occurrence of plant species in the Netherlands through species distribution models. The paper is well organized. The methods and analyses in the manuscript are robust. This is a potentially useful case study, but it is of uneven quality at present. Below are some of my suggestions.

1. I think if Fig. 1a was a map of plant species richness, it could provide more information. I’d also like to see pictures of species richness of different plant groups. Besides, could the authors please demonstrate grid cells with different proportions of natural land cover with different colors in Fig. 1b?

RE: Figure 1a was meant to show that plant data cover most if not all of the Netherlands. We now just state this in the text and moved Fig 1a to suppl. Materials (S2 Fig). Given that we analysed individual species and modelled their distribution, species richness per se is not relevant here to be depicted. We have updated Fig 1b (it is Fig 1a-c in the updated version) to reflect the gradient of natural land cover.

2. There are 1,563 species in Supplementary Table 2 while 1,544 species in Supplementary Fig. 2. Please clarify why the numbers don't tally.

RE: We ran models for 1563 species in total, but for 1554 species we could make significant response curves (i.e. increasing, decreasing and unimodal). For the other species the models were not significantly explaining their distribution. That’s why only responses of 1544 species are finally reported and were used for further analysis. 

3. In line 257-259, the authors wrote “b-d, Response curves 258 of species with increasing (each magenta line indicates one species) (b), unimodal (each blue line indicates one 259 species) (c) and decreasing (each green line indicates one species) (d) relations with NLC”. However, I think Fig. 2b showed the decreasing and Fig. 2d showed the increasing relationship with increasing NLC.

RE: They are corrected now (Fig 2, line 258-259). Thank you.

4. Please add a horizontal line to signify the line of “Representativeness = 0.0” in Fig. 3.

RE: The horizontal line indicating 0.0 value has been added to the Fig (it is Fig.4 now).

5. Most contents of the discussion were about the preferences of species of different groups for natural landscapes, but the relevant figures were placed mostly in Supporting Information. Meanwhile the authors classified species with unimodal response curves into four classes and showed several figures in the result section though with little discussion later. I think it’s better to reduce this part of the result charts and exchange with some figures regarding to preferences of different species groups.

RE: Thank you. We have moved the original Fig. S4 (it is now Fig.3), which is relevant to the first part of discussion (line 373-385), and original Fig. S13-14 (they are now Fig. 8-9) to the main text, which are relevant to the second part of discussion (line 430-451) to the main text.

6. In line 351-352, the authors wrote “Contingency Analysis revealed that threatened and rare species were more likely to occur in landscapes with NLC, NLC-F and NLC-O more than 75%”. However, Fig. 5b and 5e showed that threatened and rare species were more likely to occur in landscapes with NLC-F less than 75%. The conclusions drawn from Fig. 5 were not convincing and rigorous.

RE: The contingency Analysis actually revealed whether threatened and rare species are more likely to occur in landscape compared to not threatened and common species (see Fig 6 and line 352-355). However, if we compare the occurrence probability of threatened (or rare) species between landscapes with different NLC, more species in that category are likely to occur in landscapes with moderate NLC compared to low NLC or very high NLC as we discussed in the second paragraph in discussion.

7. The authors must have their supporting information carefully edited. There are some errors such as “rariy” in Supplementary Figure 9 and several whitespace problems.

RE: Thank you very much! I have checked the supporting information and corrected some mistakes. 

8. I suggest the authors put Supplementary Table 5 in the manuscript, not in the Supplementary information.

RE: S Table 5 has been moved to the main text (it is now Table 1).

9. Discussion 4.1 and 4.2 seems a little repetitive, the authors need to reorganize these two parts.

RE: Thanks! We have reorganized these two paragraphs.

Reviewer #3: 

1. Based on the detailed and comprehensive data, this study carried out a comprehensive study on the relationship between natural land cover types and species distribution in the Netherlands. The topics concerned by the authors have certain guiding significance for the formulation of plant diversity protection strategies and decision-making at the regional scale. However, I think the authors combined different subjects in the article, which may greatly weaken the value and significance of this study. I suggest that the author divide this research at least two papers. One is to discuss the distribution status and laws of natural land cover types from different classification perspectives. The second is the relationship between different types of gradient coverage and the distribution of species diversity of different groups. At present, because the author combines the two topics together, it is difficult to get the Internal connection between them. It is suggested that the authors should revise the paper. 

RE: Thank you for your suggestion. However, We are not exactly clear what you mean with this remark. We do not discuss the laws and status of natural land cover types in the Netherlands in contrast to what the reviewer states. Also we do not address gradients of land cover, but strictly analyse the distribution of different land cover types. We assume the reviewer may be confused in that we speak of a percentage of natural cover in a landscape. This is a summation of all the parcels in the landscape, but is not an indication of the spatial layout of the different land use types and thus is not indicative of a gradient. The other reviewers do not comment on this point and therefore we continue with the single paper that has, in our opinion, a clear message and approach.

2. Line 23: The full name of NLC should be provided. Otherwise, readers don’t know what NLC is if they don't read the text.

RE: Thank you! I have given the full definition in the abstract.

3. Line 30: The word ‘informs’ should be ‘inform’.

RE: The grammar has been corrected (line 68).

4. Lines 81-82: the words ‘natural land cover (NLC)’ should be ‘NLC’. The authors should standardize the use of abbreviations. For the words NLC, NLC –F and NLC-O, authors always used their abbreviations, or their full names, even both of them in through article. This is not in line with writing the general norms of writing.

RE: Thank you for your comment. I have standardized the use of abbreviations.

---

## [Decision Letter · Decision Letter 1]

27 Sep 2021

PONE-D-21-08410R1Importance of natural land cover for plant species’ conservation: a nationwide study in The NetherlandsPLOS ONE

Dear Dr. Pan,

Thank you for submitting your manuscript to PLOS ONE. After careful consideration, we feel that it has merit but does not fully meet PLOS ONE’s publication criteria as it currently stands. Therefore, we invite you to submit a revised version of the manuscript that addresses the points raised during the review process.

We look forward to receiving your revised manuscript.

Kind regards,

Daniel de Paiva Silva, Ph.D.

Academic Editor

PLOS ONE

Journal Requirements:

Additional Editor Comments (if provided):

Dear Pan et al.,

Congratulations, we are almost there! Please find attached the reviews from three independent reviewers. Two of them decided for the acceptance of the manuscript. The third one still raised some minor issues. In general, I think those may be corrected in the proofreading phase. Nonetheless, considering what has been raised by the reviewer, I believe we still need one final review round to solve the issues. Please be patient, because I really believe these issues will be solved very rapidly. I will grant you a one-month period. In case you need more, please let me know. DO not hesitate to submit earlier if you can.

Best regards,

Daniel Silva, PhD

Reviewers' comments:

Reviewer's Responses to Questions

**Comments to the Author**

1. If the authors have adequately addressed your comments raised in a previous round of review and you feel that this manuscript is now acceptable for publication, you may indicate that here to bypass the “Comments to the Author” section, enter your conflict of interest statement in the “Confidential to Editor” section, and submit your "Accept" recommendation.

Reviewer #1: (No Response)

Reviewer #2: All comments have been addressed

Reviewer #3: All comments have been addressed

2. Is the manuscript technically sound, and do the data support the conclusions?

Reviewer #1: Yes

Reviewer #2: Yes

Reviewer #3: Yes

3. Has the statistical analysis been performed appropriately and rigorously? 

Reviewer #1: Yes

Reviewer #2: Yes

Reviewer #3: Yes

4. Have the authors made all data underlying the findings in their manuscript fully available?

Reviewer #1: Yes

Reviewer #2: (No Response)

Reviewer #3: Yes

5. Is the manuscript presented in an intelligible fashion and written in standard English?

Reviewer #1: Yes

Reviewer #2: Yes

Reviewer #3: Yes

6. Review Comments to the Author

Reviewer #1: 1.) I am pleased the authors have removed the acronym NLC from the abstract. Issue resolved.

2.) I am pleased the authors have reworded their hypothesis to better match their methods. Issue resolved.

3.) I infer that PLOS ONE is an international journal aiming to adhere to global standards, for a global audience. I don’t perceive that informal names like “ Salix ‘Sekka’ “ are good enough by international standards regardless of whether they are accepted by the Dutch government or not. Nonetheless, now that the authors have matched the Dutch names to World Flora Online names, authorities, and species concepts – I think this does partially resolve the issue for me. Although it does open a new can of worms… So let’s look at the changes…

a) “ Salix ‘Sekka’ “ is revealed as “ Salix udensis Trautv. & C.A.Mey. “

Good. I am happy.

b) “ Arenaria leptoclados “ is revealed as “ Arenaria serpyllifolia subsp. leptoclados (Rchb.) Nyman “

It is problematic to include this taxon in this analysis as World Flora Online recognises it only as a mere subspecies of Arenaria serpyllifolia L. which itself is included in the analysis already. I would recommend that the authors remove all taxa which World Flora Online only recognises as a subspecies, to ensure methodological ‘purity’ in only performing analyses on robust species-level taxa and not ‘species-level + subspecies-level taxa’. Needlessly including taxa that are considered to be only subspecies-level by some authorities (even if not necessarily Dutch authorities), weakens the robustness of the overall analysis. If there are only a few such taxa that are questionably subspecies-only, why not simply remove them, or perform additional/complementary analyses with them removed, to demonstrate the robustness of the results?

The data that the authors themselves present in column B (“Species name World Flora Online”) of Table S2, reveals that there are eleven taxa in their dataset, including Arenaria serpyllifolia subsp. leptoclados (Rchb.) Nyman that World Flora Online recognises as subspecies only. I understand the desire to perform analyses relative to Dutch accepted species-level taxa given the Dutch geographical context, but please also (or only) perform analyses in the global/international context – that is using full, robust species-level taxa only.

Those 11 subspecies taxa are:

Arenaria serpyllifolia subsp. leptoclados (Rchb.) Nyman

Clinopodium nepeta subsp. glandulosum (Req.) Govaerts

Ranunculus peltatus subsp. baudotii (Godr.) Meikle ex C.D.K.Cook

Raphanus raphanistrum subsp. sativus (L.) Domin

Salsola kali subsp. tragus (L.) Celak.

Montia fontana subsp. chondrosperma (Fenzl) Walters

Ononis spinosa subsp. procurrens (Wallr.) Briq.

Pilosella piloselloides subsp. praealta (Gochnat) S.Bräut. & Greuter

Tripolium pannonicum subsp. tripolium (L.) Greuter

Viola tricolor subsp. curtisii (E.Forst.) Syme

Amaranthus powellii subsp. bouchonii (Thell.) Costea & Carretero

c) “ Juncus ambiguus “ is revealed as “ Juncus hybridus Brot. “

Okay.

d) “ Potamogeton mucronatus “ is revealed as “ Potamogeton friesii Rupr. “

Okay.

e) “ Festuca arenaria “ is revealed as “ Poa robusta Steud. “

Okay.

f) “ Orobanche picridis “ is revealed as “ Orobanche artemisiae-campestris Gaudin “

Okay.

g) “ Bryonia dioica “ is revealed as “ Bryonia cretica (Jacq.) Tutin “

Okay.

h) “ Carex oederi “ is revealed as “ Carex viridula Michx. “

Okay.

i) “ Carex ovalis “ is revealed as “ Carex leporina L. “

Okay.

j) “ Rubus fruticosus “ is revealed as “ Rubus vestitus Weihe “

Okay.

k) “ Tilia x vulgaris “ is revealed as “ Tilia × europaea L. “

Okay.

l) “ Centaurea montana “ is revealed as “ Cyanus montanus (L.) Hill “

Okay.

With the addition of mappings to World Flora Online names and authorities, the data now becomes less cryptic, and more reproducible. Thank you.

4a) I apologise to authors for mistakenly alleging they did not include fern & clubmoss species in this analysis in my first-round review, I can see now they indeed do have some fern and clubmoss species. It would be nice to include a Class-level indication for each taxon in Table S2 e.g. “Lycopodiopsida” , “Polypodiopsida” , “Pinopsida” , “Magnoliopsida” to better illustrate the diversity of the data. If not Class-level then relative to the Order-level of the APG III system?

b) with respect to _land cover_ and entirely aquatic plant species such as the duckweed species included in this work

I think I may have also made a mistake here in assuming that water could not be a type of ‘land cover’. I now see that by normal parlance water is an accepted type of land cover.

Nevertheless, I can’t help but feel that strictly aquatic-only species might have considerably different properties to strictly terrestrial-only species. Does aquatic non-natural land cover even exist? Isn’t all aquatic land cover ‘natural’ or ‘semi-natural’, and thus aren’t aquatic-only species a possible source of bias in this analysis (assuming non-natural aquatic land cover does not exist).

Relatedly how do aquatic-only species fit into testing the sub-hypothesis of “forests” versus “open habitats”. “Forests” and “open habitats” seem to me to be defined on the basis of terrestriality – I think the aquatic-only species should be excluded from this sub-hypothesis analysis, as I wouldn’t expect to find duckweed in areas of predominantly “forest” land-cover. I also note (~line 108) that cells containing >10% of open water were excluded from the analysis – isn’t that excluding the cells where duckweed and other aquatic-only species are quite likely to be found? Does this really not bias this sub-hypothesis analysis to be including aquatic-only vascular plant species in it?

5.) I am satisfied with the explanation given in the reply to reviewers, but please explain this in the manuscript too.

6.) I am pleased that the authors have matched Dutch names to World Flora Online names. As I mention in 3b) this reveals some further issues.

7.) Thank you for including this extra data

8.) Hmmm… I guess it’s not strictly necessary to add this information but it would certainly be helpful

9.) Why not just upload the code to Zenodo? Zenodo just not charge, it is also a not for profit organisation. Whilst I understand the data has some proprietary protections on it imposed by NDFF, your code has no such encumberment.

Thank you.

Reviewer #2: (No Response)

Reviewer #3: (No Response)

7. PLOS authors have the option to publish the peer review history of their article (what does this mean?). If published, this will include your full peer review and any attached files.

Reviewer #1: **Yes: **Ross Mounce

Reviewer #2: **Yes: **Yue Xu

Reviewer #3: No

---

## [Author Response · Author response to Decision Letter 1]

11 Oct 2021

Journal Requirements:

RE: Thank you. We have checked it and all references have been cited correctly.

 

Reviewer #1: 

Comment 3. I infer that PLOS ONE is an international journal aiming to adhere to global standards, for a global audience. I don’t perceive that informal names like “ Salix ‘Sekka’ “ are good enough by international standards regardless of whether they are accepted by the Dutch government or not. Nonetheless, now that the authors have matched the Dutch names to World Flora Online names, authorities, and species concepts – I think this does partially resolve the issue for me. Although it does open a new can of worms… So let’s look at the changes…

b) “ Arenaria leptoclados “ is revealed as “ Arenaria serpyllifolia subsp. leptoclados (Rchb.) Nyman “

It is problematic to include this taxon in this analysis as World Flora Online recognises it only as a mere subspecies of Arenaria serpyllifolia L. which itself is included in the analysis already. I would recommend that the authors remove all taxa which World Flora Online only recognises as a subspecies, to ensure methodological ‘purity’ in only performing analyses on robust species-level taxa and not ‘species-level + subspecies-level taxa’. Needlessly including taxa that are considered to be only subspecies-level by some authorities (even if not necessarily Dutch authorities), weakens the robustness of the overall analysis. If there are only a few such taxa that are questionably subspecies-only, why not simply remove them, or perform additional/complementary analyses with them removed, to demonstrate the robustness of the results?

The data that the authors themselves present in column B (“Species name World Flora Online”) of Table S2, reveals that there are eleven taxa in their dataset, including Arenaria serpyllifolia subsp. leptoclados (Rchb.) Nyman that World Flora Online recognises as subspecies only. I understand the desire to perform analyses relative to Dutch accepted species-level taxa given the Dutch geographical context, but please also (or only) perform analyses in the global/international context – that is using full, robust species-level taxa only.

Those 11 subspecies taxa are:

Arenaria serpyllifolia subsp. leptoclados (Rchb.) Nyman

Clinopodium nepeta subsp. glandulosum (Req.) Govaerts

Ranunculus peltatus subsp. baudotii (Godr.) Meikle ex C.D.K.Cook

Raphanus raphanistrum subsp. sativus (L.) Domin

Salsola kali subsp. tragus (L.) Celak.

Montia fontana subsp. chondrosperma (Fenzl) Walters

Ononis spinosa subsp. procurrens (Wallr.) Briq.

Pilosella piloselloides subsp. praealta (Gochnat) S.Bräut. & Greuter

Tripolium pannonicum subsp. tripolium (L.) Greuter

Viola tricolor subsp. curtisii (E.Forst.) Syme

Amaranthus powellii subsp. bouchonii (Thell.) Costea & Carretero

With the addition of mappings to World Flora Online names and authorities, the data now becomes less cryptic, and more reproducible. Thank you.

RE: Thank you. We have removed these 11 species and reconducted analysis in the international context. Results have been updated in the main text.

Comment 4. 

4a): I apologise to authors for mistakenly alleging they did not include fern & clubmoss species in this analysis in my first-round review, I can see now they indeed do have some fern and clubmoss species. It would be nice to include a Class-level indication for each taxon in Table S2 e.g. “Lycopodiopsida” , “Polypodiopsida” , “Pinopsida” , “Magnoliopsida” to better illustrate the diversity of the data. If not Class-level then relative to the Order-level of the APG III system?

RE: Thank you for your suggestion! We have given the group, order and family according to the latest APG IV in the updated S2 Table.

4b): with respect to _land cover_ and entirely aquatic plant species such as the duckweed species included in this work

I think I may have also made a mistake here in assuming that water could not be a type of ‘land cover’. I now see that by normal parlance water is an accepted type of land cover.

(1) Nevertheless, I can’t help but feel that strictly aquatic-only species might have considerably different properties to strictly terrestrial-only species. Does aquatic non-natural land cover even exist? Isn’t all aquatic land cover ‘natural’ or ‘semi-natural’, and thus aren’t aquatic-only species a possible source of bias in this analysis (assuming non-natural aquatic land cover does not exist).

RE: In the Netherlands water is ubiquitous and much of the land is neither natural nor semi-natural. Small parcels of aquatic non-natural land cover occur as part of urban green and grey (small ditches, ponds, etc), crop and pasture (small ditches, etc), in fact in all land uses. Thus explaining the results.

(2) Relatedly how do aquatic-only species fit into testing the sub-hypothesis of “forests” versus “open habitats”. “Forests” and “open habitats” seem to me to be defined on the basis of terrestriality – I think the aquatic-only species should be excluded from this sub-hypothesis analysis, as I wouldn’t expect to find duckweed in areas of predominantly “forest” land-cover. 

RE: Indeed as you can see from table S2 the preference of duckweed spp (e.g. Lemna and Wolffia, which was mentioned by the reviewer in the first round of review) for forest is 0 or very low. By the way, the reviewer may not realize that categorizing a location as forested versus open is not based on terrestriality, it is based on the major vegetation structure (=land use type). This does not mean that 100% of a location is dense forest or completely open, some locations have both, but one as the dominant type. Also many forests in the Netherlands include small streams, ponds or ditches which are habitats for all kinds of aquatic spp (incl. for example the duckweed spp that tend to occur in nutrient rich standing water). There are very few locations in the Netherlands without water.

(3) I also note (~line 108) that cells containing >10% of open water were excluded from the analysis – isn’t that excluding the cells where duckweed and other aquatic-only species are quite likely to be found? Does this really not bias this sub-hypothesis analysis to be including aquatic-only vascular plant species in it?

RE: 10% water is 10 ha, which means a decent amount of open water. Normally, no (or few) duckweeds and other aquatic-only species live in those open waters as we explained in the last response letter. It means those removed grid cells are not highly relevant to most species. In addition, as we explained, the Netherlands consists mostly of water. Most habitats where aquatic plants grow have small streams, ditches, ponds, marsh or wetland areas rather than larger areas of open water. In this way, removing open water (>10%) grid cells will not bias this hypothesis.

Comment 5. I am satisfied with the explanation given in the reply to reviewers, but please explain this in the manuscript too.

RE: We have indicated it in the main text (Line 179-181). Thank you!

Comment 6. I am pleased that the authors have matched Dutch names to World Flora Online names. As I mention in 3b) this reveals some further issues.

RE: We have addressed the problem in comment 3b).

Comment 8. Hmmm… I guess it’s not strictly necessary to add this information but it would certainly be helpful

RE: A higher-level taxonomy for each species has been added to the S2 Table.

Comment 9. Why not just upload the code to Zenodo? Zenodo just not charge, it is also a not for profit organisation. Whilst I understand the data has some proprietary protections on it imposed by NDFF, your code has no such encumberment.

RE: Thank you for your suggestion. We feel our response, making code available upon request is sufficient and fulfills the journal’s policy.

---

## [Decision Letter · Decision Letter 2]

18 Oct 2021

Importance of natural land cover for plant species’ conservation: a nationwide study in The Netherlands

PONE-D-21-08410R2

Dear Dr. Pan,

We’re pleased to inform you that your manuscript has been judged scientifically suitable for publication and will be formally accepted for publication once it meets all outstanding technical requirements.

Kind regards,

Daniel de Paiva Silva, Ph.D.

Academic Editor

PLOS ONE

Additional Editor Comments (optional):

Reviewers' comments:

Reviewer's Responses to Questions

**Comments to the Author**

1. If the authors have adequately addressed your comments raised in a previous round of review and you feel that this manuscript is now acceptable for publication, you may indicate that here to bypass the “Comments to the Author” section, enter your conflict of interest statement in the “Confidential to Editor” section, and submit your "Accept" recommendation.

Reviewer #1: All comments have been addressed

2. Is the manuscript technically sound, and do the data support the conclusions?

Reviewer #1: Yes

3. Has the statistical analysis been performed appropriately and rigorously? 

Reviewer #1: Yes

4. Have the authors made all data underlying the findings in their manuscript fully available?

Reviewer #1: Yes

5. Is the manuscript presented in an intelligible fashion and written in standard English?

Reviewer #1: Yes

6. Review Comments to the Author

Reviewer #1: Thank you for patiently explaining it to me. I have nothing more to say.

7. PLOS authors have the option to publish the peer review history of their article (what does this mean?). If published, this will include your full peer review and any attached files.

Reviewer #1: **Yes: **Ross Mounce

---

## [Editor Report · Acceptance letter]

22 Oct 2021

PONE-D-21-08410R2 

Importance of natural land cover for plant species’ conservation: a nationwide study in The Netherlands 

Dear Dr. Pan:

I'm pleased to inform you that your manuscript has been deemed suitable for publication in PLOS ONE. Congratulations! Your manuscript is now with our production department. 

Kind regards, 

on behalf of

Dr. Daniel de Paiva Silva 

Academic Editor

PLOS ONE